# Rethinking LLM-as-a-Judge: Representation-as-a-Judge with Small Language Models via Semantic Capacity Asymmetry

**Zhuochun Li**[1,2,†], **Yong Zhang**[1,†], **Ming Li**[3], **Yuelyu Ji**[2], **Yiming Zeng**[4], **Ning Cheng**[1,*],
**Yun Zhu**[1], **Yanmeng Wang**[1], **Shaojun Wang**[1], **Jing Xiao**[1], **Daqing He**[2]

[1]Ping An Technology (Shenzhen) Co., Ltd. [2]University of Pittsburgh

[3]University of Maryland, College Park [4]University of Connecticut

```
{zhl163,yuj49,dah44}@pitt.edu
{zhangyong203,chengning211}@pingan.com.cn
{minglii}@umd.edu  {yiming.zeng}@uconn.edu
```

## Abstract

Large language models (LLMs) are widely used as reference-free evaluators via prompting, but this "LLM-as-a-Judge" paradigm is costly, opaque, and sensitive to prompt design. In this work, we investigate whether smaller models can serve as efficient evaluators by leveraging internal representations instead of surface generation. We uncover a consistent empirical pattern: small LMs, despite with weak generative ability, encode rich evaluative signals in their hidden states. This motivates us to propose the Semantic Capacity Asymmetry Hypothesis: evaluation requires significantly less semantic capacity than generation and can be grounded in intermediate representations, suggesting that evaluation does not necessarily need to rely on large-scale generative models but can instead leverage latent features from smaller ones. Our findings motivate a paradigm shift from *LLM-as-a-Judge* to *Representation-as-a-Judge*, a decoding-free evaluation strategy that probes internal model structure rather than relying on prompted output. We instantiate this paradigm through **INSPECTOR**, a probing-based framework that predicts aspect-level evaluation scores from small model representations. Experiments on reasoning benchmarks (GSM8K, MATH, GPQA) show that INSPECTOR substantially outperforms prompting-based small LMs and closely approximates full LLM judges, while offering a more efficient, reliable, and interpretable alternative for scalable evaluation. The code and data are available at: https://github.com/zhuochunli/Representation-as-a-judge

## 1 Introduction

Large language models (LLMs) have demonstrated remarkable capabilities in generation, reasoning, and alignment tasks (Achiam et al., 2023; Touvron et al., 2023). A growing number of works leverage the paradigm of *LLM-as-a-Judge*, wherein powerful LLMs are prompted to assess the quality of generated outputs without access to ground-truth references (Chang et al., 2024; Prasad et al., 2023). This approach has achieved strong empirical results in reference-free evaluation across domains such as summarization and complex reasoning (He et al., 2023; Zhang et al., 2024).

However, this prompt-based evaluation paradigm has important limitations. First, it requires autoregressive decoding, making it computationally expensive even for single-point evaluations. Second, it relies on large proprietary models (e.g., GPT-4), whose internal mechanisms remain opaque and unverifiable. Lastly, its effectiveness depends heavily on prompt engineering, raising concerns about reproducibility, robustness, and scaling (Polo et al., 2024; Voronov et al., 2024; Mizrahi et al., 2024).

---

† Equal contribution.
∗ Corresponding author.
Work was done during Zhuochun Li's internship at Ping An Technology.

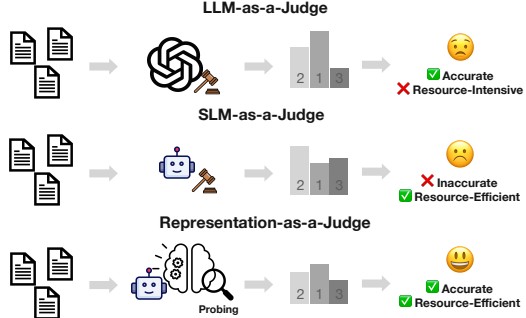

Figure 1: Illustration of **Representation-as-a-Judge.**

A natural goal is to use smaller open-source LMs as evaluators, as they offer a more lightweight and accessible alternative to large proprietary models. However, when prompted directly, their evaluation performance is poor and highly inconsistent compared to large LLMs. Prior work (Li et al., 2024a; Waldis et al., 2024) has shown that small models, despite weaker generation, often possess semantic competence comparable to large models. This suggests that their poor evaluation performance may stem from limitations in surface generation rather than a fundamental lack of understanding. Building on this, we ask a more granular question: *do small models encode evaluation-relevant signals in their internal representations, even when generation is poor?*

This question represents a broader insight: the ability to *evaluate* may place lower demands on semantic capacity than the ability to *generate*. Even when generation fails, compact internal representations may already encode the features needed for judgment. We therefore formalize the **Semantic Capacity Asymmetry Hypothesis**: *The semantic capacity required for accurate evaluation is significantly lower than that required for generation. Evaluation can be grounded in compressed internal representations of small language models, even when generation requires full-model decoding.*

Building on this hypothesis, we advance an alternative perspective on evaluation: **Representation-as-a-Judge**. Rather than relying on prompted text generation, evaluative signals can be extracted directly from the latent structure. This addresses key bottlenecks of prompt-based evaluation and opens the door to lightweight, interpretable, and scalable systems.

To explore this perspective, we introduce **INSPECTOR** ( INternal Signal Probing and EvaluaTion Of Representations), a probing-based framework for reference-free evaluation. Given a (prompt, response) pair, INSPECTOR performs the following steps: (1) obtain aspect-level evaluation scores from a strong LLM judge across multiple aspects (e.g., logicality, fluency, consistency); (2) input the same evaluation prompt to a small LM and extract internal representations; (3) identify informative layers and train lightweight classifiers to approximate evaluation scores using latent embeddings.

Empirically, we validate this framework on reasoning benchmarks such as GSM8K, MATH, and GPQA, and find that internal representations from small models (e.g., 1.7B) achieve high predictive performance, substantially outperforming prompting-based baselines and in many cases approaching the fidelity of full-scale LLM judges. Furthermore, we show that classifiers trained with this method can be used to filter noisy reasoning data, leading to measurable gains in downstream supervised fine-tuning (Xu et al., 2024; Albalak et al., 2024; Li et al., 2024b).

**Our contributions are threefold:**

- We identify and analyze a consistent empirical phenomenon: small LMs, despite weak generation, encode evaluation-relevant signals in their internal representations.

- We formalize this insight as the **Semantic Capacity Asymmetry Hypothesis**, positing that evaluation requires less semantic capacity than generation and can be grounded in intermediate representations.

- We advance a new perspective, **Representation-as-a-Judge**, and instantiate it through our probing-based framework, **INSPECTOR**, demonstrating high-fidelity evaluation and efficient data filtering for reasoning tasks. To the best of our knowledge, this is the first work to investigate LLM probing for evaluation tasks.

Figure 2: Overview of our proposed **INSPECTOR**. We freeze the small LMs and probe their representations, training only a lightweight probing classifier to fit the proprietary LLM evaluations.

## 2 RELATED WORK

Our work builds on two research directions: (i) data evaluation methods, particularly the emerging *LLM-as-a-Judge* paradigm, and (ii) probing techniques for analyzing LLM representations. Below we summarize progress in each line and position our contribution.

**LLM Evaluation** Recent studies in NLP have introduced reference-based evaluation metrics such as BERTScore (Zhang et al., 2019) and BARTScore (Yuan et al., 2021), which leverage pretrained language models to better align with human judgments. ROSCOE further develops unsupervised, reference-free metrics that outperform traditional n-gram methods like ROUGE (Lin, 2004) and BLEU (Papineni et al., 2002), as well as several model-based metrics. With the emergence of LLMs, many works have begun to treat LLMs themselves as evaluators (Zheng et al., 2023; Liu et al., 2023; Li et al., 2024c), identifying and filtering high-quality data samples. A variety of techniques enhance this LLM-as-a-Judge paradigm: Chain-of-Thought reasoning (Wei et al., 2022) to improve decision quality, RECEVAL (Prasad et al., 2023) to assess reasoning chains via correctness and informativeness, and SOCREVAL (He et al., 2023) to let LLMs generate their own answers before evaluation. However, these methods often rely on prompt engineering and proprietary LLMs, limiting both their reliability and interpretability. Alternatively, we propose to probe and analyze critical internal representations of small LMs, training lightweight classifiers to approximate state-of-the-art evaluations from powerful LLMs. This yields a more efficient and explainable approach to data evaluation.

**Probing LLM Representations** There is a growing interest in probing the internal representations of LLMs to uncover interpretable features. Early work by Shi et al. (2016) introduced probing by training a logistic regression classifier on top of machine translation encoders to study the extent of syntactic information. Building on this idea, later studies investigate more complex linguistic phenomena. For example, Starace et al. (2023) examine how linguistic categories such as part-of-speech (POS) and dependency relations are jointly encoded across layers of LLMs, revealing shared and hierarchical structures in their representations. Beyond linguistic categories, recent research has explored whether LLMs capture higher-level abstractions of world knowledge and state. Zhang et al. (2025) propose Sentinel, which probes decoder attention in small proxy LLMs to extract relevance signals for context compression, framing probing as a lightweight understanding task rather than a linguistic diagnostic. More works further probe the internal representations of world states in Transformer models when processing game scripts and embodied sequences (Li et al., 2023; Jin & Rinard, 2024; Di Palma et al., 2025).

In contrast to prior work, which mainly seeks to understand what knowledge LLMs encode, we leverage probing in a new direction: extracting critical internal representations that are predictive of evaluation quality. To the best of our knowledge, this is the first work to bridge probing with the LLM-as-a-Judge paradigm, enabling evaluation that is both more efficient and more interpretable.

## 3 METHODOLOGY

The **Semantic Capacity Asymmetry Hypothesis** suggests that evaluation can be grounded in compact intermediate representations of small LMs, without requiring full decoding. To operationalize and test this idea, we propose REPRESENTATION-AS-A-JUDGE: a new evaluation paradigm that directly probes latent semantic structure rather than relying on surface-level outputs. We instantiate this paradigm through **INSPECTOR**. Illustrated in Figure2, it consists of three components: LLMs evaluation annotation, small LMs probing, and building probing classifiers, which we will discuss in Sec. 4.1, 4.2, and 4.3, respectively.

## 3.1 LLMs EVALUATION ANNOTATION

Following established rubric definitions from prior work, including ROSCOE Golovneva et al. (2022) and Socreval He et al. (2023), we adopt five widely used evaluation aspects $\mathcal{K}$ to assess the quality of reference-free rationales, with the corresponding few-shot prompts provided in Appendix K:

- **Semantic Consistency**: Do the solution steps and final answer remain faithful to the problem facts (no invented events, omitted givens, or unstated assumptions)?

- **Logicality**: Does each inference and arithmetic step follow valid rules and correctly apply operations?

- **Informativeness**: Does the rationale include the essential steps and intermediate calculations needed to verify the final answer?

- **Fluency**: Is the text grammatical, clear, and easy to follow, with proper punctuation, sentence flow, notation, and presentation?

- **Factuality**: Are the claims, facts, evidence, references, and concrete assertions in the rationale factually correct and supported?

Let $x$ denote a task instruction and $y$ the corresponding model-generated response. We first employ a medium-scale language model $\mathcal{M}_{med}$ (10-50B) to generate responses for all benchmark queries. Unlike state-of-the-art LLMs $\mathcal{M}_{large}$ (50–100B+), $\mathcal{M}_{med}$ is less powerful, which is desirable because it produces more diverse evaluation distributions, containing both good and bad quality samples, across different aspects, thereby facilitating the subsequent probing process. This will construct the response dataset $D = \{(x_i, y_i)\}_{i=1}^{N}$, where $x_i$ is a benchmark question and $y_i = \mathcal{M}_{med}(x_i)$ is the corresponding response. For each evaluation aspect $k \in \mathcal{K}$, we construct an aspect-specific instruction $\mathcal{I}_k$ and integrate $(x, y)$ into it, yielding $\mathcal{I}_k(x, y)$ as the evaluation prompt. We then query a state-of-the-art LLM $\mathcal{M}_{large}$ to obtain evaluation scores along each aspect. For a given sample $(x, y)$, we prompt $\mathcal{M}_{large}$ with $\mathcal{I}_k(x, y)$ and obtain a scalar score $s_k \in [1, 5]$:

$$s_{i,k} = \mathcal{M}_{large}\big(\mathcal{I}_k(x_i, y_i)\big).$$

Collecting these pairs yields the probing dataset

$$D_{prob} = \bigcup_{k \in \mathcal{K}} \big\{ (\mathcal{I}_k(x_i, y_i), \, s_{i,k}) \big\}_{i=1}^{N}. \tag{1}$$

To avoid bias toward over-represented quality levels, we construct a balanced probing dataset for each aspect: we first determine the minimum number of samples among the five score levels (1–5), and then randomly downsample the other levels to this size. This ensures that the multiclass probing tasks are balanced across labels and not dominated by frequent scores.

Specifically, original scores (1–5) are treated as **multiclass classification tasks**, while responses with scores higher than threshold $\tau$ are labeled high quality and the rest low quality, forming a simpler **binary classification task**. Since these aspects have been validated and the effectiveness of powerful LLMs has been extensively studied in prior work, we treat the resulting scores as gold labels for our subsequent LM probing experiments.

## 3.2 SMALL LMs PROBING

We use probing to test whether small LMs encode linearly recoverable evaluative cues in their hidden states, with the probe's minimal capacity ensuring that any predictive signal reflects the model's own semantics. In our tasks, we observe that prompt-based evaluation inference from small LMs $\mathcal{M}_{small}$ (0-10B) yields a large gap compared to the gold scores from $\mathcal{M}_{large}$, primarily due to differences in model capacity. To address this, instead of relying solely on the final decoded text, we analyze internal representations layer by layer to identify those most predictive of the gold evaluation scores. Specifically, we convert each prompt $\mathcal{I}_k(x_i, y_i) \in D_{prob}$ into a collection of per-layer, pooled representation features, and fit these features with simple, cross-validated linear probes. More explanations could be found in Appendix B.

**Extraction and pooling.** For a sample $i$, let $S_i$ denote the sequence length and $t = 1, ..., S_i$ index tokens, we input prompt $\mathcal{I}_k(x_i, y_i)$ and run $\mathcal{M}_{\text{small}}$ in evaluation mode, obtaining per-layer $\ell$ hidden states $\mathbf{H}_i^{(\ell)}$ and attention weights $\mathbf{A}_i^{(\ell)}$. From $\mathbf{H}_i^{(\ell)}$ we compute a small but expressive set of pooled vectors: mean, last, min, max, and concat. These pooling variants capture complementary token-level and global signals across layers. For example:

$$
\begin{aligned}
\mathbf{r}_{i,\text{mean}}^{(\ell)} &= \tfrac{1}{S_i} \sum_{t=1}^{S_i} \mathbf{H}_i^{(\ell)}[t, :] \in \mathbb{R}^d \\
\mathbf{r}_{i,\text{last}}^{(\ell)} &= \mathbf{H}_i^{(\ell)}[t_i^*, :] \in \mathbb{R}^d, \quad t_i^* = \max\{t : \text{token } t \text{ non-pad}\}
\end{aligned}
\tag{2}
$$

**Attention and statistical features.** For each head $h$ of layer $\ell$, we compute an attention-entropy statistic $e_{i,h}^{(\ell)}$ and compress them into the number of attention heads $R$ per layer:

$$
\mu_i^{(\ell)} = \frac{1}{R} \sum_h e_{i,h}^{(\ell)}, \quad \sigma_i^{(\ell)} = \text{std}_h\big(e_{i,h}^{(\ell)}\big), \quad \max_h e_{i,h}^{(\ell)}.
\tag{3}
$$

For each pooled vector $\mathbf{r}$ we also compute compact statistics, including its norm $\text{norm}(\mathbf{r})$, variance $\text{var}(\mathbf{r})$, and entropy $E(\mathbf{r})$.

**Feature assembly.** For each layer $\ell$ and pooling type $p \in \{\text{mean,last,min,max,concat}\}$, we construct a feature matrix by concatenating multiple components along the feature dimension:

$$
X^{(\ell,p)} = \big[\, \text{PCA}_d(\mathbf{r}_{i,p}^{(\ell)}) \mid \underbrace{[\text{norm}, \text{var}, E(\cdot)]}_{\text{statistics}} \mid [\mu_i^{(\ell)}, \sigma_i^{(\ell)}, \max_h e_{i,h}^{(\ell)}] \,\big]_{i=1}^N
\tag{4}
$$

where $\text{PCA}_d$ denotes an optional dimensionality-reduction to $d$ components. All transforms that depend on the data (imputer, PCA, scaler) are applied inside a cross-validation pipeline to avoid information leakage.

**Probing and layer ranking.** We treat gold labels $s_{i,k} \in \{1, \ldots, 5\}$ from $D_{\text{prob}}$ as both (i) a multiclass prediction target $y_{i,k}^{\text{multi}} = s_{i,k}$ and (ii) a binary target $y_{i,k}^{\text{bin}} = \mathbb{I}[s_{i,k} \geq \tau]$ with threshold $\tau$ (high vs. low quality). For each $X^{(\ell,p)}$ we fit a logistic probe and evaluate generalization with stratified cross-validation. Based on the fitting results, we rank layer–pool–feature configurations by a chosen criterion (binary or multiclass performance) for different downstream tasks.

The probing stage produces: (1) a ranked list of layer–pool–feature configurations with cross-validated predictive performance, (2) per-layer progression plots that visualize where evaluative signal accumulates inside $\mathcal{M}_{\text{small}}$. These results inform the subsequent section, where we build final probing classifiers using the selected features.

### 3.3 BUILDING PROBING CLASSIFIERS

The probing stage produces a ranked list of layer–pool–feature configurations with predictive performance. In this section we describe how we use that ranked list to assemble multi-layer feature sets, train final probing classifiers, and select an optimal evaluator.

**From ranked configurations to candidate layer sets.** Let $\pi$ denote the ranked list of layer–pool–feature tuples produced after the previous probing stage. We take the **Top-K** unique layers in $\pi$ and start from the highest-ranked single layer, iteratively adding the next-ranked layer only if the addition improves performance.

**Feature concatenation for multi-layer probes.** For a chosen subset of layers $S = \{\ell_1, \ldots, \ell_{|S|}\}$ and a pooling method $p$, we construct a multi-layer feature matrix by horizontally concatenating the per-layer features defined in Eq. equation 4. Concretely, for sample $i$ we form the concatenated feature vector

$$
\tilde{\mathbf{x}}_i^{(S,p)} = \big[\, \mathbf{r}_{i,p}^{(\ell_1)}; \mathbf{r}_{i,p}^{(\ell_2)}; \ldots; \mathbf{r}_{i,p}^{(\ell_{|S|})} \,\big] \in \mathbb{R}^{|S| \cdot d_p},
\tag{5}
$$

where $d_p$ is the dimensionality of the pooled vector for pooling $p$. If attention summaries are included, we append the per-layer attention summaries to form the final feature vector. The assembled dataset for $N$ examples is $\widetilde{X}^{(S,p)} = [\tilde{\mathbf{x}}_1^{(S,p)}, \ldots, \tilde{\mathbf{x}}_N^{(S,p)}]^\top$.

**Classifier training and selection.** For each candidate feature assembly $\widetilde{X}^{(S,p)}$, we train a family of simple, interpretable classifiers to predict both the multiclass target $y^{\text{multi}}$ and the binary target $y^{\text{bin}}$. We select the final probing classifier by maximizing a task-specific performance criterion over the candidate set $\mathcal{S}$ and probe families. Formally, letting $\mathcal{C}$ denote the set of classifier hyperparameterizations tested, we choose

$$(S^\star, p^\star, \theta^\star) = \underset{(S,p,\text{clf})}{\text{argmax}} \ \bar{a}_\gamma^{(S,p,\text{clf})}, \quad (S, p, \text{clf}) \in \mathcal{S} \times \mathcal{P} \times \mathcal{C}. \tag{6}$$

where $\gamma \in \{\text{bin}, \text{multi}\}$ denotes the classification task type (binary vs. multiclass), and $\theta^\star$ are the learned classifier parameters for the selected configuration. In case of ties, we prefer the configuration with smaller $\sigma$ (more stable performance) and with fewer layers.

This search yields a compact, high-performing probing classifier that uses a small subset of layers identified by the probing stage, which is tuned for robust generalization via cross-validation and grid search. It can serve as an efficient surrogate evaluator approximating $\mathcal{M}_{\text{large}}$ judgments while remaining orders of magnitude cheaper at inference time.

## 4 EXPERIMENTS

### 4.1 EXPERIMENTAL SETUP

**Datasets.** To prove the effectiveness on reasoning evaluation tasks, we picked three popular benchmarks in Mathematics & Science Tasks: GSM8K (Cobbe et al., 2021), MATH (Hendrycks et al., 2021), and GPQA (Rein et al., 2024). Datasets statistics are shown in Appendix C.

**Models.** In our experiments, we select Llama-3-8B-Instruct (Dubey et al., 2024) as $\mathcal{M}_{\text{med}}$ for response generation, as it achieves reasonable performance on the chosen benchmarks while producing diverse evaluation scores. For LLM-based evaluation annotation, we employ DeepSeek-V3 (Liu et al., 2024) as $\mathcal{M}_{\text{large}}$ owing to its strong reasoning ability and relatively low cost. To validate the effectiveness of our pipeline across different small LMs, we consider Qwen3-0.6B (Yang et al., 2025), Qwen3-1.7B (Yang et al., 2025), Llama-3.2-1B-Instruct (Dubey et al., 2024), and Llama-3.1-8B-Instruct (Dubey et al., 2024) as $\mathcal{M}_{\text{small}}$.

**Baselines.** To the best of our knowledge, this is the first study to investigate LLM probing for evaluation tasks, and therefore related work is limited. Accordingly, we compare our approach against three baselines: (1) direct prompt-based inference on small LMs, (2) fine-tuning small LMs (Qwen3-0.6B), and (3) RoBERTa (Liu et al., 2019) on probing datasets to fit the evaluation scores.

**Implementation Details.** We set the score threshold $\tau = 4$, labeling samples with scores $\geq 4$ as high-quality and those with scores $< 4$ as low-quality. For dimensionality reduction, we apply PCA with $d = 50$ in the probing process. We use $K = 5$ for Top-$K$ aggregation and experiment with a variety of classifiers, including Logistic Regression, Random Forests, small Multilayer Perceptrons (MLPs), and linear Support Vector Machines (SVMs). Since all pooling, layer, and classifier variants operate on cached hidden representations, exploring these configurations incurs negligible computational overhead. All test results are reported on the zero-shot prompt and weighted average F1 score. Additional implementations are provided in Appendix D.

### 4.2 PROBING CLASSIFIERS RESULTS

We present the main results of our optimal probing classifiers across Mathematics & Science benchmarks in Figure 3. Explicit statistics for the probing datasets and the numbers of main results can be found in Appendix E, F. Notably, our strong results are obtained by only training on small probing datasets (typically fewer than 100 samples per score) due to the downsampling strategy (3.1).

**Probing is much more effective than prompt-based inference.** Figure 3 shows substantial improvements of our methods over baselines, with average F1 scores increasing by more than 20% on most tasks. This suggests that **poor outputs from LLMs do not necessarily imply that the models lack the underlying knowledge to solve the task**. Instead, **crucial information may be**

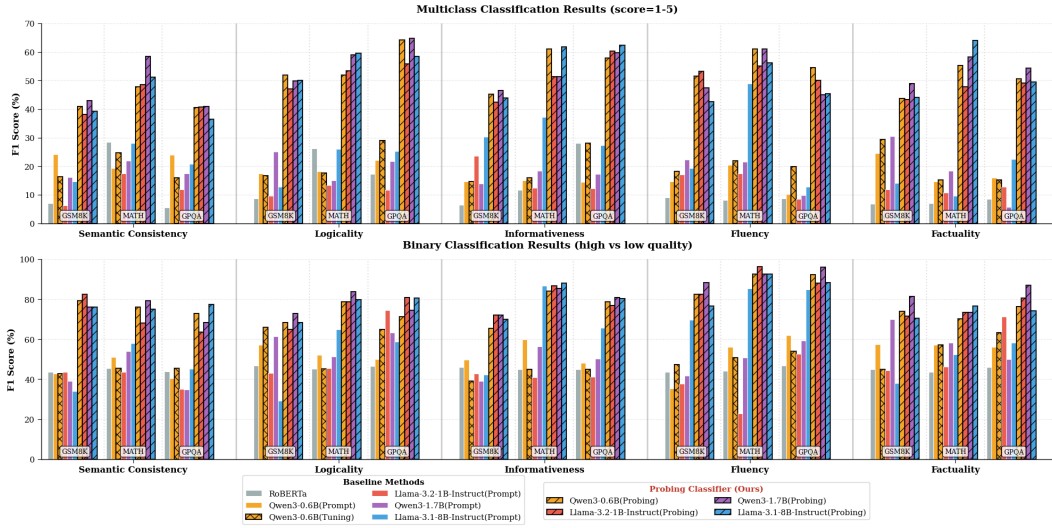

Figure 3: Weighted average F1 score (%) across reasoning benchmarks with multiclass classification and binary classification tasks. Our probing method (colored bars with hatch marks) significantly outperforms prompting on the same models (colored bars without hatch marks) across all tasks.

**already embedded in the internal representations, but remains hidden after the final decoding process**. Probing allows us to uncover and leverage these latent understandings, whereas direct text generation can introduce noise and degrade human-interpretable performance. Importantly, the effectiveness of our approach is consistently observed across all evaluation aspects and for different sizes of $\mathcal{M}_{small}$ across multiple model families, including results after fine-tuning. This provides a strong indication that our method can be applied to more general scenarios.

**Larger LLMs do not necessarily provide stronger evaluation, either through inference or probing.** Although Llama-3.1-8B-Instruct achieves the best results on several tasks, Qwen3-1.7B still outperforms it on certain benchmarks despite being much smaller. Within the same model family, both the Qwen3 and Llama3 series demonstrate that larger models do not consistently surpass their smaller counterparts across all aspects. For instance, on MATH, Qwen3-0.6B outperforms Qwen3-1.7B in prompt-based inference for logicality (18.18% vs. 15.06%), and Llama-3.2-1B-Instruct surpasses Llama-3.1-8B-Instruct in binary probing for fluency (96.32% vs. 92.65%). These results highlight that different models can exhibit distinct strengths across evaluation aspects, cautioning against a blind reliance on scaling laws.

**Probing classifiers for binary classification serve as highly reliable data filters.** While the performance of probing classifiers on multiclass prediction remains modest (approximately 50–60%), this is expected given the difficulty of approximating a $\mathcal{M}_{large}$ that is hundreds of times larger with a $\mathcal{M}_{small}$. In contrast, binary classification performance reaches 80–90%, making it sufficiently reliable to function as a reference-free coarse filtering mechanism. This capability is particularly valuable for common NLP applications such as curating high-quality data for supervised fine-tuning, where the filter can efficiently separate high and low quality samples during initial screening and thereby reduce the cost of further fine-grained annotation.

## 5  ANALYSIS

### 5.1  ABLATION STUDY OF POOLING AND CLASSIFIER METHODS

For each probing result in Figure 3, we don't report the combinations of various pooling methods ($\mathbf{r}_i^{(\ell)}$ in Equation 2) and classifiers (clf in Equation 6). To better illustrate the effect of different pooling and classifier choices, we conduct an ablation study on the Informativeness aspect of the

MATH dataset, using Qwen3-0.6B and Llama-3.2-1B-Instruct on binary classification, with results shown in Figure 4.

The results demonstrate that mean pooling consistently outperforms other strategies, which is intuitive since averaging preserves critical information while producing compact feature representations. Among classifiers, Logistic Regression achieves the best results. With limited data and potentially noisy labels from LLMs scoring, calibrated probabilities and regularization from Logistic Regression can lead to a better overall F1 score. These patterns match findings from recent studies: Kantamneni et al. (2025) show that complex probes often do not strongly outperform simpler pooling + linear classifiers; Lee et al. (2023) demonstrates that averaging over all token vectors can beat using only CLS or last token embeddings. Finally, these findings are consistent with the detailed results shown in Table 11, where top-performing configurations most often involve mean pooling combined with Logistic Regression classifiers.

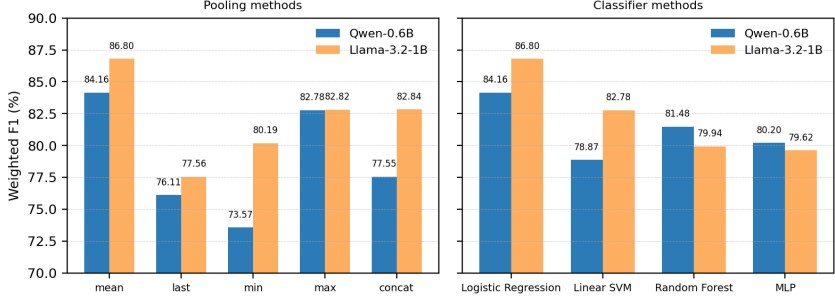

Figure 4: Ablation study of different pooling and classifier methods on binary classification tasks. Left: logistic regression fixed. Right: mean pooling fixed.

## 5.2 DATA FILTERING AND SUPERVISED FINE-TUNING (SFT)

To assess the effectiveness of our data filtering approach, we apply the trained probing classifiers of Qwen3-1.7B in a knowledge distillation setup, with Llama-3-8B-Instruct (Dubey et al., 2024) as the teacher and Llama-2-7B-Chat (Touvron et al., 2023) as the student. For each benchmark, the classifiers assign binary scores (0 or 1) to every response across five aspects, which are then summed to yield a total score between 0 and 5. Responses are ranked by this score from high to low to construct the training set, and the student is trained on progressively larger subsets in 10% increments. We directly compare our probing-based filtering against two baselines: (i) the gold DeepSeek-V3 filtering and (ii) random filtering. The result curves are summarized in Figure 5.

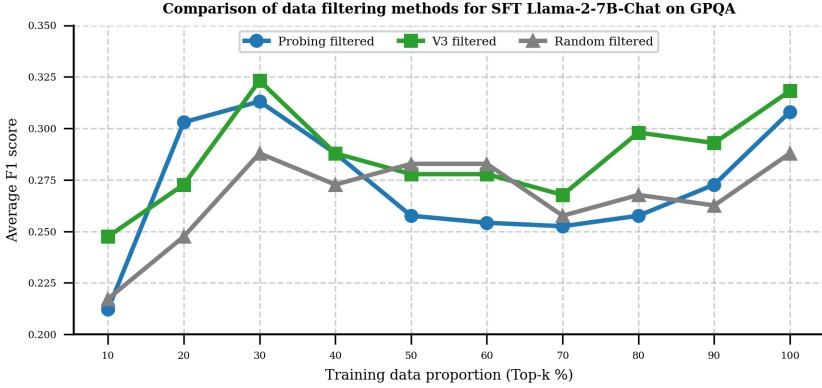

Figure 5: Llama-2-7B-Chat SFT performance under different data filtering methods (probing classifier, DeepSeek-V3, and random) with incremental training subsets.

Our findings can be summarized as follows: (1) filtering training data with our probing classifiers yields SFT performance comparable to using powerful LLM as the filter, indicating that our approach can approximate LLM-level data quality judgments; (2) both probing and DeepSeek-V3 consistently outperform random filtering, despite occasional mid-range fluctuations where random

sampling may incidentally capture higher-quality data, reinforcing the importance of quality-aware data selection for downstream training; and (3) the observed **up–down–up trend** suggests that initial gains derive from training on high-quality data, performance then declines as lower-quality data is introduced, and subsequently recovers once the volume of training data becomes sufficiently large. This supports claims in prior studies (Sajith & Kathala, 2024; Iyer et al., 2024): **data quality plays the primary role in low-resource settings, whereas data quantity may become a dominant factor as training data scales**.

## 6 SEMANTIC CAPACITY ASYMMETRY IN EVALUATIVE SIGNALS

### 6.1 EVALUATIVE SIGNALS IN INTERMEDIATE REPRESENTATIONS

To better understand why small models can support accurate evaluation, we analyze the internal representations used in our probing classifiers (§4). Specifically, we examine how evaluative signals vary across layers and what types of features most effectively capture them. We present representative results in Figure 6, with additional analysis provided in Appendix I.

**(1) Representations Encode Strong Evaluative Signals.** Across reasoning datasets, we observe that hidden representations show substantial correlation with evaluation scores from the strong LLM judge. These signals are especially strong in mid-to-upper layers, indicating that evaluative information is embedded throughout intermediate layers, rather than being restricted to the output stage.

**(2) Evaluative Features Reside in Structured Feature Subspaces.** Probing feature spaces derived from PCA-based subspaces often reveals evaluative signals more effectively than scalar or attention-derived features. These observations suggest that evaluative signals are present across structured feature subspaces, with PCA-based projections revealing them more clearly than simpler features.

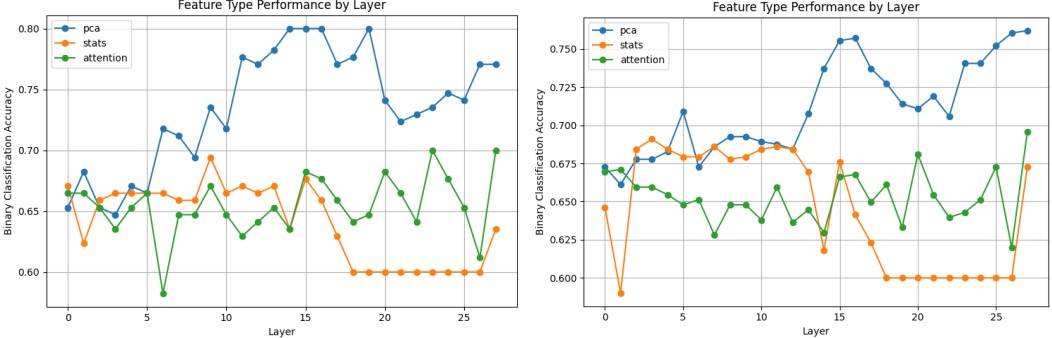

Figure 6: Layer-wise probing accuracy for **Factuality** (left) and **Semantic Consistency** (right) using **Qwen3-1.7B** on the **MATH** dataset. PCA features perform best, peaking in upper layers.

### 6.2 THE SEMANTIC CAPACITY ASYMMETRY HYPOTHESIS

These patterns observed above support the hypothesis that accurate evaluation requires much less capacity than generation and can rely on intermediate representations. This reflects an intrinsic asymmetry between the tasks: generation involves discourse planning and long dependencies that demand substantial capacity and often require full decoding, while evaluation focuses on identifying inconsistencies or content errors, which are already accessible in intermediate model states.

Prior work has shown that internal representations often encode much richer and more reliable semantic information than what is reflected in surface outputs. Probing studies demonstrate that hidden states capture fine-grained linguistic and semantic structure (Waldis et al., 2024; Rogers et al., 2020). Complementing this, latent-knowledge analyses reveal that task-relevant information can remain present in intermediate representations even when generated text is unreliable or intentionally misled (Kadavath et al., 2022; Burns et al., 2023; Mallen et al., 2023). Building on these insights, we show that intermediate representations not only encode general semantic structure but also contain evaluation-relevant signals that can be directly leveraged for effective reference-free assessment.

# 7 CONCLUSION

Despite suboptimal generation, small LMs retain strong evaluative signals in their internal representations. We validate the Semantic Capacity Asymmetry Hypothesis and introduce INSPECTOR, a probing-based pipeline that extracts high-fidelity judgments from these latents. Experiments on various reasoning benchmarks show the capability of our method, suggesting that Representation-as-a-Judge can serve as a scalable and interpretable solution for evaluation and data curation.

## REPRODUCIBILITY STATEMENT

All datasets and models used in our experiments are publicly accessible. We also provide additional details in the Appendix, including dataset statistics, model parameters, and training hyperparameters. We believe that the information provided is sufficient to reproduce our methods and results. Furthermore, we released all data and code in the public repository: `https://github.com/zhuochunli/Representation-as-a-judge`.

## USE OF LARGE LANGUAGE MODELS (LLMS)

We used large language models (LLMs) only for minor text polishing, such as grammar and phrasing. All ideas, experiments, analyses, and discussions were conducted solely by the authors. The LLM did not contribute to the design and interpretation of our research.

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

## A  LIMITATIONS

While the proposed method demonstrates competitive performance compared to other baselines, we acknowledge that there are still potential limitations:

- Following prior work, we adopt five evaluation aspects and construct corresponding prompt templates for our experiments. However, in the absence of standardized evaluation criteria, our chosen definitions may not represent the optimal formulation for all tasks. Furthermore, we observe that the difficulty of the selected evaluation aspects varies. For instance, fluency appears to be the easiest, as the majority of responses consistently receive high scores relative to other aspects. This suggests the need for further investigation into the design and selection of evaluation aspects.

- Although we conducted experiments on various datasets, future work can explore more general reasoning fields, such as commonsense and code generation tasks.

- We only use the DeepSeek-V3 as our rating model, due to its cost and availability. However, this may cause evaluation bias and affect our downstream probing classifier training process. Exploring diverse rating LLMs from different organizations could be a valuable direction for future research.

## B  SMALL LMS PROBING DETAILS

**Notation and extraction.**   For a sample $i$, we input prompt $\mathcal{I}_k(x_i, y_i)$ and run $\mathcal{M}_{\text{small}}$ in evaluation mode, obtaining

$$\mathbf{H}_i^{(\ell)} \in \mathbb{R}^{S_i \times d}, \qquad \mathbf{A}_i^{(\ell)} \in \mathbb{R}^{R \times S_i \times S_i},$$

for layers $\ell = 1, \ldots, L$. Here $S_i$ is the token length, $d$ the hidden dimension, and $R$ the number of attention heads.

**Pooling and representation.**   From $\mathbf{H}_i^{(\ell)}$ we compute a small but expressive set of pooled vectors. These pooling variants capture complementary token-level and global signals across layers:

$$\mathbf{r}_{i,\text{mean}}^{(\ell)} = \frac{1}{S_i} \sum_{t=1}^{S_i} \mathbf{H}_i^{(\ell)}[t,:] \in \mathbb{R}^d,$$

$$\mathbf{r}_{i,\text{last}}^{(\ell)} = \mathbf{H}_i^{(\ell)}[t_i^*,:] \in \mathbb{R}^d, \quad t_i^* = \max\{t : \text{token } t \text{ non-pad}\},$$

$$\mathbf{r}_{i,\text{min}}^{(\ell)} = \min_t \mathbf{H}_i^{(\ell)}[t,:], \quad \mathbf{r}_{i,\text{max}}^{(\ell)} = \max_t \mathbf{H}_i^{(\ell)}[t,:],$$

$$\mathbf{r}_{i,\text{concat}}^{(\ell)} = \left[\mathbf{r}_{i,\text{min}}^{(\ell)}; \mathbf{r}_{i,\text{max}}^{(\ell)}; \mathbf{r}_{i,\text{mean}}^{(\ell)}\right] \in \mathbb{R}^{3d}.$$

**Attention and statistical features.** For each head $h$ of layer $\ell$ we compute an attention-entropy statistic (with small constant $\epsilon > 0$ for numerical stability):

$$e_{i,h}^{(\ell)} = -\frac{1}{S_i} \sum_{s=1}^{S_i} \sum_{t=1}^{S_i} A_{i,h,s,t}^{(\ell)} \log\left(A_{i,h,s,t}^{(\ell)} + \epsilon\right).$$

We compress head-wise entropies into low-dimensional summaries per layer:

$$\mu_i^{(\ell)} = \frac{1}{R} \sum_h e_{i,h}^{(\ell)}, \quad \sigma_i^{(\ell)} = \text{std}_h\left(e_{i,h}^{(\ell)}\right), \quad \max_h e_{i,h}^{(\ell)}.$$

For each pooled vector $\mathbf{r}$ we also compute compact statistics:

$$\text{norm}(\mathbf{r}) = \|\mathbf{r}\|_2, \quad \text{var}(\mathbf{r}) = \text{Var}(\mathbf{r}), \quad E(\mathbf{r}) = -\sum_m \text{softmax}(\mathbf{r})_m \log \text{softmax}(\mathbf{r})_m,$$

**Feature assembly.** For each layer $\ell$ and pooling type $p \in \{\text{mean,last,min,max,concat}\}$ we assemble candidate feature matrices:

$$X^{(\ell,p)} = \big[\ \text{PCA}_d(\mathbf{r}_{i,p}^{(\ell)}) \ \big|\ \underbrace{[\text{norm, var, } E(\cdot)]}_{\text{statistics}} \ \big|\ [\mu_i^{(\ell)}, \sigma_i^{(\ell)}, \max_h e_{i,h}^{(\ell)}]\big]_{i=1}^N$$

where $\text{PCA}_d$ denotes an optional dimensionality-reduction to $d$ components. All transforms that depend on the data (imputer, PCA, scaler) are applied inside a cross-validation pipeline to avoid information leakage.

**Probing and layer ranking.** We treat gold labels $s_{i,k} \in \{1, \ldots, 5\}$ from $D_{\text{prob}}$ as both (i) a multiclass prediction target $y_{i,k}^{\text{multi}} = s_{i,k}$ and (ii) a binary target $y_{i,k}^{\text{bin}} = \mathbb{I}[\,s_{i,k} \geq \tau\,]$ with threshold $\tau$ (high vs. low quality). For each $X^{(\ell,p)}$ we fit a logistic probe (linear logistic regression; one-vs-rest for multiclass) and evaluate generalization with stratified cross-validation.

Each candidate probe yields a performance tuple

$$\left(\bar{a}_{\text{bin}}^{(\ell,p)},\ \sigma_{\text{bin}}^{(\ell,p)},\ \bar{a}_{\text{multi}}^{(\ell,p)},\ \sigma_{\text{multi}}^{(\ell,p)}\right)$$

where $\bar{a}_{\text{bin}}^{(\ell,p)}$ and $\bar{a}_{\text{multi}}^{(\ell,p)}$ denote the mean accuracies of the binary and multiclass probes, respectively, and $\sigma_{\text{bin}}^{(\ell,p)}, \sigma_{\text{multi}}^{(\ell,p)}$ their corresponding standard deviations across cross-validation folds. We rank layer–pool–feature configurations by a chosen criterion (binary or multiclass accuracy mean) for different downstream tasks.

## C  DATASETS STATISTICS

We download datasets GSM8K and GPQA from Huggingface, MATH from their official project website: `https://github.com/hendrycks/math`. GSM8K dataset is split according to the official original split ratio. We use the official training set for Math and MATH-500 for the test set due to its high representation and low cost. Since there is no official train/test split for GPQA, we use gpqa_main and gpqa_extended as the training set, gpqa_diamond as the test set. Table 1 shows the statistics of all datasets.

| Dataset | Type | #Train | #Test |
|---------|------|--------|-------|
| GSM8K | Mathematics | 7473 | 1319 |
| MATH | Mathematics | 7500 | 500 |
| GPQA | Science | 994 | 198 |

Table 1: Dataset statistics.

## D  IMPLEMENTATION DETAILS

Primary experiments are conducted on eight NVIDIA Quadro RTX 8000 and eight NVIDIA RTX A6000 GPUs.

## D.1 SMALL LMS PARAMETERS

The parameter settings for $\mathcal{M}_{\text{small}}$: Qwen3-0.6B (Yang et al., 2025), Qwen3-1.7B (Yang et al., 2025), Llama-3.2-1B-Instruct (Dubey et al., 2024), and Llama-3.1-8B-Instruct (Dubey et al., 2024).

| Parameter | Value |
|---|---|
| temperature | 0.6 |
| max new tokens | 256 |
| do sample | True |
| output_hidden_states | True |
| output_attentions | True |
| torch_dtype | float16 |
| attn_implementation | eager |

Table 2: Small LMs parameter settings.

## D.2 MEDIUM LMS PARAMETERS

The parameter settings for $\mathcal{M}_{\text{med}}$: Llama-3-8B-Instruct (Dubey et al., 2024).

| Parameter | Value |
|---|---|
| temperature | 0 |
| max new tokens | 512 |
| do sample | True |
| torch_dtype | float16 |
| top_p | 1.0 |

Table 3: Medium LM parameter settings.

## D.3 LARGE LMS PARAMETERS

DeepSeek-V3 (Liu et al., 2024) is required by the official API: `https://api-docs.deepseek.com`.

| Parameter | Value |
|---|---|
| temperature | 0 |
| max new tokens | 2048 |
| do sample | True |
| model | deepseek-chat |
| response_format | {'type': 'json_object'} |

Table 4: Large LM parameter settings.

## D.4 MODEL TUNING HYPERPARAMETER

We list the training hyperparameters for Llama-2-7B-Chat (Touvron et al., 2023) in Section 6.4.

## D.5 PROBING CLASSIFIER DETAILS

We train several probing classifiers on internal features to fit gold scores. All classifiers are implemented in scikit-learn pipelines with StandardScaler for feature normalization. We perform hyperparameter search via 5-fold cross-validation using macro F1 as the scoring metric. Table 6 below summarizes each classifier's fixed parameters and the hyperparameter ranges considered in the grid search.

| Hyperparameter | Value |
|---|:---:|
| epoch | 8 |
| batch size | 8 |
| learning rate | 1e-4 |
| warmup_steps | 100 |
| max seq length | 1024 |
| gradient accumulation steps | 4 |
| lora_r | 16 |
| lora_alpha | 32 |
| lora_dropout | 0.1 |
| target_modules | ["q_proj", "v_proj"] |

Table 5: Student LM training hyperparameter settings.

| Classifier | Fixed Parameters | Hyperparameter Grid |
|---|---|---|
| Logistic Regression | `max_iter=2000,` `class_weight="balanced",` `solver="lbfgs"` | `C:` [0.001,0.01,0.1,1]; `penalty:` ["l2"] |
| Random Forest | `class_weight="balanced",` `random_state=42` | `n_estimators:` [100,300,500]; `max_depth:` [None,10,20]; `min_samples_leaf:` [1,2,5] |
| Multi-layer Perceptron | `hidden_layer_sizes=(256,128),` `activation="relu",` `alpha=1e-4,` `learning_rate_init=1e-3,` `max_iter=1000,` `early_stopping=True,` `random_state=42` | `alpha:` [1e-4,1e-3,1e-2]; `learning_rate_init:` [1e-4,1e-3,1e-2]; `hidden_layer_sizes:` [(200,100),(100,),(200,100,50)] |
| Linear SVM | `kernel="linear",` `class_weight="balanced",` `probability=True,` `random_state=42` | C: [0.001,0.01,0.1,1,10,100] |

Table 6: Probing classifiers including fixed parameters and grid search ranges for hyperparameter tuning.

# E PROBING DATASETS STATISTICS

As explained in Section 4.1, we first determine the minimum number of samples $n$ among the five score levels (1–5), and then randomly downsample the other levels to this size. This ensures that the multiclass probing tasks are balanced across labels and not dominated by frequent scores. Thus, after downsampling, $D_{prob}$ contains a total of $5 \times n$ samples, and we split the dataset into training and test sets with an 80:20 ratio. In the following tables, we mark the minimum number $n$ of each aspect in **bold**. These score distributions can reflect the difficulty evaluation levels among various dataset questions and aspects.

$D_{prob}$ statistics for GSM8K:

$D_{prob}$ statistics for MATH:

$D_{prob}$ statistics for GPQA:

|  | #score=1 | #score=2 | #score=3 | #score=4 | #score=5 |
|---|---|---|---|---|---|
| Semantic Consistency | 184 | 491 | 369 | **34** | 6435 |
| Logicality | 118 | 842 | 174 | **26** | 6353 |
| Informativeness | **46** | 81 | 111 | 49 | 7226 |
| Fluency | **17** | 23 | 70 | 86 | 7317 |
| Factuality | 377 | 300 | 479 | **85** | 6272 |

Table 7: Number of LLM-judge scores across five aspects of probing datasets on GSM8K.

|  | #score=1 | #score=2 | #score=3 | #score=4 | #score=5 |
|---|---|---|---|---|---|
| Semantic Consistency | 441 | 467 | 1879 | **121** | 4575 |
| Logicality | 224 | 1288 | 1604 | **98** | 4269 |
| Informativeness | **75** | 180 | 748 | 2062 | 4418 |
| Fluency | 132 | **27** | 746 | 2857 | 3721 |
| Factuality | 609 | 945 | 1818 | **34** | 4077 |

Table 8: Number of LLM-judge scores across five aspects of probing datasets on MATH.

|  | #score=1 | #score=2 | #score=3 | #score=4 | #score=5 |
|---|---|---|---|---|---|
| Semantic Consistency | 257 | 529 | 60 | **22** | 125 |
| Logicality | 267 | 463 | 145 | **31** | 87 |
| Informativeness | 197 | 164 | 327 | 163 | **142** |
| Fluency | 135 | **26** | 68 | 210 | 554 |
| Factuality | 306 | 475 | 119 | **46** | 47 |

Table 9: Number of LLM-judge scores across five aspects of probing datasets on GPQA.

# F DETAILED MAIN RESULTS

We show the specific numbers of results in Table 10.

| Method | Semantic Consistency | | | Logicality | | | Informativeness | | | Fluency | | | Factuality | | |
|---|---|---|---|---|---|---|---|---|---|---|---|---|---|---|---|
| | GSM8K | MATH | GPQA | GSM8K | MATH | GPQA | GSM8K | MATH | GPQA | GSM8K | MATH | GPQA | GSM8K | MATH | GPQA |
| **Multiclass Classification (score=1-5)** | | | | | | | | | | | | | | | |
| RoBERTa (Liu et al., 2019) | 7.03 | 28.34 | 5.59 | 8.65 | 26.18 | 17.18 | 6.40 | 11.65 | 28.14 | 8.96 | 8.08 | 8.65 | 6.73 | 7.03 | 8.50 |
| **Qwen3-0.6B** (Yang et al., 2025) | | | | | | | | | | | | | | | |
| Prompt Inference | 24.21 | 19.33 | 23.92 | 17.46 | 18.18 | 22.12 | 14.62 | 15.09 | 14.53 | 14.66 | 20.47 | 10.10 | 24.50 | 14.65 | 15.96 |
| Tuning | 16.38 | 24.76 | 16.00 | 16.64 | 17.68 | 29.03 | 14.63 | 16.01 | 28.14 | 18.18 | 21.91 | 19.84 | 29.35 | 15.23 | 15.16 |
| Probing Classifier | 40.92 | 47.73 | 40.48 | **51.93** | 52.00 | 64.23 | 45.18 | 61.07 | 57.97 | 51.46 | 60.97 | **54.60** | 43.80 | 55.18 | 50.67 |
| **Llama-3.2-1B-Instruct** (Dubey et al., 2024) | | | | | | | | | | | | | | | |
| Prompt Inference | 6.33 | 17.40 | 11.82 | 9.55 | 13.32 | 11.73 | 23.54 | 12.34 | 12.12 | 17.11 | 17.52 | 8.45 | 11.76 | 10.74 | 12.83 |
| Probing Classifier | 38.04 | 48.47 | 40.69 | 47.16 | 53.47 | 55.81 | 42.33 | 51.27 | 60.34 | **53.29** | 55.16 | 50.13 | 43.29 | 47.80 | 49.21 |
| **Qwen3-1.7B** (Yang et al., 2025) | | | | | | | | | | | | | | | |
| Prompt Inference | 16.18 | 21.98 | 17.42 | 25.00 | 15.06 | 21.81 | 13.88 | 18.34 | 17.27 | 22.27 | 21.45 | 9.77 | 30.46 | 18.40 | 5.66 |
| Probing Classifier | **42.98** | **58.45** | **40.93** | 49.86 | 59.05 | **64.79** | **46.51** | 51.42 | 59.79 | 47.39 | **60.97** | 45.02 | **48.86** | 58.34 | **54.37** |
| **Llama-3.1-8B-Instruct** (Dubey et al., 2024) | | | | | | | | | | | | | | | |
| Prompt Inference | 14.72 | 27.97 | 20.78 | 12.75 | 26.04 | 25.34 | 30.20 | 37.21 | 27.35 | 19.33 | 48.94 | 12.69 | 14.07 | 9.63 | 22.53 |
| Probing Classifier | 39.31 | 51.10 | 36.41 | 50.00 | **59.63** | 58.42 | 43.85 | **61.78** | **62.28** | 42.56 | 56.15 | 45.36 | 44.06 | **63.94** | 49.56 |
| **Binary-Classification (high vs low quality)** | | | | | | | | | | | | | | | |
| RoBERTa (Liu et al., 2019) | 43.57 | 45.40 | 43.90 | 46.89 | 45.25 | 46.58 | 46.06 | 45.00 | 44.83 | 43.57 | 44.10 | 46.89 | 45.00 | 43.57 | 46.06 |
| **Qwen3-0.6B** (Yang et al., 2025) | | | | | | | | | | | | | | | |
| Prompt Inference | 42.71 | 51.13 | 40.27 | 57.22 | 51.99 | 49.93 | 49.71 | 59.74 | 48.21 | 35.29 | 56.08 | 61.95 | 57.41 | 57.26 | 55.99 |
| Tuning | 42.71 | 45.40 | 45.45 | 65.85 | 45.25 | 64.75 | 38.98 | 45.00 | 44.83 | 47.43 | 50.73 | 53.85 | 45.00 | 57.26 | 63.21 |
| Probing Classifier | 79.28 | 76.07 | 72.95 | 68.43 | 78.80 | 71.16 | 65.48 | 84.16 | 78.73 | 82.35 | 92.65 | 92.31 | 73.80 | 70.16 | 76.37 |
| **Llama-3.2-1B-Instruct** (Dubey et al., 2024) | | | | | | | | | | | | | | | |
| Prompt Inference | 43.57 | 43.48 | 35.12 | 43.08 | 45.45 | 74.35 | 42.79 | 41.03 | 41.31 | 37.65 | 22.93 | 52.75 | 44.44 | 46.28 | 71.17 |
| Probing Classifier | **82.35** | 68.07 | 63.64 | 65.00 | 78.79 | **80.85** | 72.08 | 86.80 | 76.93 | 82.35 | **96.32** | 87.94 | 71.42 | 73.36 | 80.52 |
| **Qwen3-1.7B** (Yang et al., 2025) | | | | | | | | | | | | | | | |
| Prompt Inference | 38.95 | 53.88 | 34.66 | 61.54 | 51.34 | 63.28 | 38.94 | 56.28 | 50.26 | 41.58 | 50.73 | 59.17 | 69.85 | 58.22 | 49.90 |
| Probing Classifier | 76.13 | **79.21** | 68.38 | **72.78** | **83.67** | 74.47 | **72.08** | 85.48 | **80.96** | **88.32** | 92.65 | **96.11** | **81.26** | 73.36 | **87.11** |
| **Llama-3.1-8B-Instruct** (Dubey et al., 2024) | | | | | | | | | | | | | | | |
| Prompt Inference | 34.02 | 58.01 | 45.10 | 29.22 | 64.80 | 58.81 | 42.32 | 86.67 | 65.63 | 69.56 | 85.27 | 84.80 | 38.02 | 52.26 | 58.12 |
| Probing Classifier | 76.13 | 75.11 | **77.42** | 68.43 | 79.77 | 80.65 | 69.91 | **88.12** | 80.41 | 76.63 | 92.65 | 88.33 | 70.51 | **76.47** | 74.21 |

Table 10: Average F1 score (%) across various reasoning benchmarks with multiclass classification and binary classification tasks. The best performance among different classification tasks in each benchmark is marked in **bold**.

We also report the specific layers, pooling, and classifier methods that achieve the best probing performance, as summarized in Table 10, across different benchmarks in Table 11.

| | GSM8K | MATH | GPQA |
|---|---|---|---|
| | **Multiclass Classification (score=1-5)** | | |
| Semantic Consistency | Qwen3-1.7B, layers=[14,25], pool="mean", LR | Qwen3-1.7B, layers=[15], pool="mean", LS | Qwen3-1.7B, layers=[2], pool="mean", LS |
| Logicality | Qwen3-0.6B, layers=[17], pool="mean", LR | Llama-3.1-8B-Instruct, layers=[17], pool="last", LR | Qwen3-1.7B, layers=[13], pool="mean", LS |
| Informativeness | Qwen3-1.7B, layers=[16,17], pool="mean", LS | Llama-3.1-8B-Instruct, layers=[10], pool="mean", LS | Llama-3.1-8B-Instruct, layers=[12], pool="mean", LS |
| Fluency | Llama-3.2-1B-Instruct, layers=[1], pool="last", LS | Qwen3-1.7B, layers=[18], pool="mean", LS | Qwen3-0.6B, layers=[18], pool="last", LR |
| Factuality | Qwen3-1.7B, layers=[14,18], pool="mean", LR | Llama-3.1-8B-Instruct, layers=[21], pool="mean", LR | Qwen3-1.7B, layers=[15], pool="last", LR |
| | **Binary-Classification (high vs low quality)** | | |
| Semantic Consistency | Llama-3.2-1B-Instruct, layers=[3], pool="last", LS | Qwen3-1.7B, layers=[15], pool="mean", LR | Llama-3.1-8B-Instruct, layers=[16], pool="last", LR |
| Logicality | Qwen3-1.7B, layers=[24], pool="last", LR | Qwen3-1.7B, layers=[18], pool="last", LR | Llama-3.2-1B-Instruct, layers=[3], pool="last", LR |
| Informativeness | Qwen3-1.7B, layers=[21], pool="mean", LS | Llama-3.1-8B-Instruct, layers=[20], pool="last", LS | Qwen3-1.7B, layers=[17], pool="mean", LR |
| Fluency | Qwen3-1.7B, layers=[1,11], pool="last", LS | Llama-3.2-1B-Instruct, layers=[13], pool="last", LR | Qwen3-1.7B, layers=[14], pool="mean", LR |
| Factuality | Qwen3-1.7B, layers=[16], pool="mean", LR | Llama-3.1-8B-Instruct, layers=[28], pool="mean", LR | Qwen3-1.7B, layers=[18], pool="last", LS |

Table 11: Detailed layers, pooling and classifiers selection for best probing performances. "pool" denotes different pooling methods in Equation 2. "LR" denotes Logistic Regression and "LS" denotes linear SVM.

## G  OUT-OF-DISTRIBUTION (OOD) PROBING

To evaluate the generalization abilities of our probing methods on Out-of-Distribution (OOD) data, we conducted experiments using one mathematical reasoning dataset as the training set and another dataset as the test set. Table 12 highlights the probing performance with Qwen3-1.7B as the $\mathcal{M}_{small}$ in OOD scenarios. Our analysis reveals two complementary patterns. First, multiclass (1–5) probing demonstrates limited transferability, with F1 scores dropping to approximately 10–25%, suggesting that fine-grained score prediction is dataset-specific. In particular, transferring from more challenging datasets (MATH) to simpler ones (GSM8K) proves more difficult than the reverse. Second, binary probing demonstrates substantially greater robustness under distribution shift: out-of-distribution F1 scores range from ∼35–62%, comparable to in-distribution results in Table 10. These findings suggest that PCA-based linear probing reliably captures coarse, domain-general quality signals, whereas fine-grained distinctions are too difficult to transfer. Consequently, for claims of OOD robustness, we emphasize binary evaluations and recommend controlling probe capacity when reporting cross-dataset transfer.

| Method | Semantic Consistency | | Logicality | | Informativeness | | Fluency | | Factuality | |
|---|---|---|---|---|---|---|---|---|---|---|
| | GSM8K | MATH | GSM8K | MATH | GSM8K | MATH | GSM8K | MATH | GSM8K | MATH |
| | **Multiclass Classification (score=1-5)** | | | | | | | | | |
| **Qwen3-1.7B** Probing Classifier | 10.31 | 14.26 | 10.19 | 23.50 | 16.47 | 11.65 | 13.35 | 11.60 | 16.63 | 27.57 |
| | **Binary-Classification (high vs low quality)** | | | | | | | | | |
| **Qwen3-1.7B** Probing Classifier | 35.76 | 61.04 | 27.20 | 62.90 | 60.39 | 45.00 | 41.58 | 23.59 | 36.70 | 62.19 |

Table 12: Average F1 score (%) of test results across mathematics reasoning benchmarks in Out-of-Distribution (OOD) scenarios. Specifically, we conducted experiments by training on GSM8K and testing on MATH, as well as training on MATH and testing on GSM8K.

## H  EXTENDED RESULTS ON OPEN-ENDED BENCHMARK

While our primary focus is on reasoning evaluation, we additionally evaluated our approach on the open-ended generation benchmark AlpacaEval 2.0 (Dubois et al., 2024), using the same experimental settings as the three reasoning benchmarks in Table 10. The results of AlpacaEval 2.0 are shown in Table 13. The results further supports the effectiveness and robustness of our approach on a broader range of domains and tasks, including reference-free open-ended benchmarks.

| Method | Semantic Consistency | Logicality | Informativeness | Fluency | Factuality |
|---|---|---|---|---|---|
| **Multiclass Classification (score=1-5)** | | | | | |
| RoBERTa (Liu et al., 2019) | 7.56 | 8.08 | 8.32 | 11.69 | 8.08 |
| **Qwen3-0.6B** (Yang et al., 2025) | | | | | |
| Prompt Inference | 22.40 | 11.11 | 16.94 | 11.82 | 8.89 |
| Tuning | 14.29 | 11.11 | 9.93 | 10.61 | 12.70 |
| Probing Classifier | 36.84 | 42.59 | 46.71 | 61.21 | 23.33 |
| **Llama-3.2-1B-Instruct** (Dubey et al., 2024) | | | | | |
| Prompt Inference | 12.50 | 13.33 | 8.37 | 16.97 | 0.20 |
| Probing Classifier | 35.71 | 55.56 | 46.08 | 61.21 | 28.57 |
| **Qwen3-1.7B** (Yang et al., 2025) | | | | | |
| Prompt Inference | 7.14 | 20.11 | 22.70 | 25.19 | 20.00 |
| Probing Classifier | 35.12 | 45.93 | 42.01 | 65.10 | 23.70 |
| **Llama-3.1-8B-Instruct** (Dubey et al., 2024) | | | | | |
| Prompt Inference | 14.92 | 17.99 | 18.34 | 23.38 | 14.29 |
| Probing Classifier | 34.69 | 53.70 | 45.90 | 57.32 | 22.96 |
| **Binary-Classification (high vs low quality)** | | | | | |
| RoBERTa (Liu et al., 2019) | 41.56 | 39.68 | 46.58 | 49.49 | 39.68 |
| **Qwen3-0.6B** (Yang et al., 2025) | | | | | |
| Prompt Inference | 55.24 | 43.00 | 49.93 | 46.36 | 44.44 |
| Tuning | 35.24 | 43.00 | 45.41 | 28.48 | 27.78 |
| Probing Classifier | 61.42 | 55.56 | 62.76 | 91.06 | 55.56 |
| **Llama-3.2-1B-Instruct** (Dubey et al., 2024) | | | | | |
| Prompt Inference | 49.20 | 34.19 | 59.26 | 64.24 | 39.68 |
| Probing Classifier | 67.14 | 70.42 | 67.44 | 81.82 | 65.80 |
| **Qwen3-1.7B** (Yang et al., 2025) | | | | | |
| Prompt Inference | 50.24 | 65.80 | 45.03 | 45.45 | 55.56 |
| Probing Classifier | 67.14 | 75.93 | 70.71 | 91.06 | 77.78 |
| **Llama-3.1-8B-Instruct** (Dubey et al., 2024) | | | | | |
| Prompt Inference | 39.77 | 48.15 | 58.34 | 62.42 | 27.35 |
| Probing Classifier | 70.16 | 77.78 | 69.96 | 82.12 | 64.07 |

Table 13: Average F1 score (%) on AlpacaEval 2.0 with multiclass classification and binary classification tasks.

# I  ADDITIONAL ANALYSIS OF EVALUATIVE SIGNALS

To complement the main findings in Figure 6, we include detailed layer-wise analyses for additional evaluation dimensions: *Factuality*, *Informativeness*, *Logicality*, and *Semantic Consistency*. These results are based on **Qwen3-1.7B** evaluated on the **MATH** dataset. Each figure reports probing accuracy across layers using three feature types: PCA-projected embeddings, statistical summaries, and attention-derived vectors.

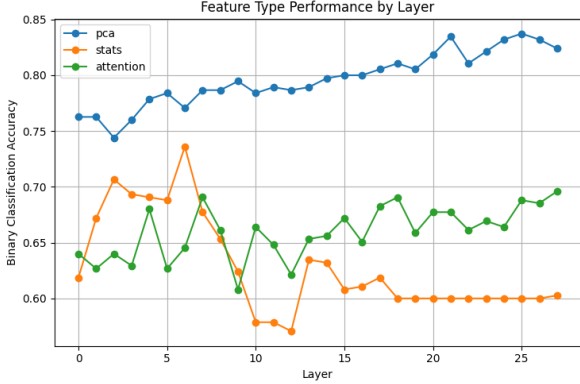

Figure 7: Layer-wise diagnostic results for **Informativeness**. Peak correlation appears near Layer 25. PCA accuracy increases steadily, while stats features decline in upper layers.

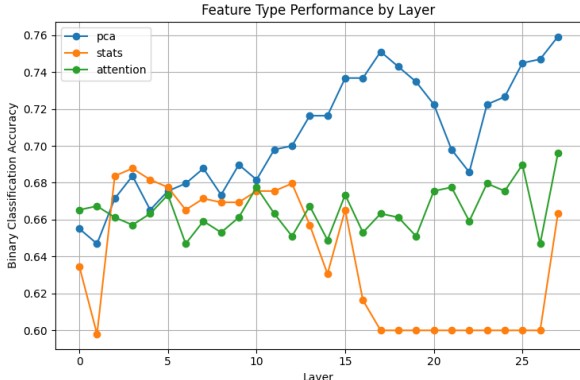

Figure 8: Layer-wise diagnostic results for **Logicality**. Signals concentrate around Layers 17 and 27. PCA features show rising accuracy toward the upper layers.

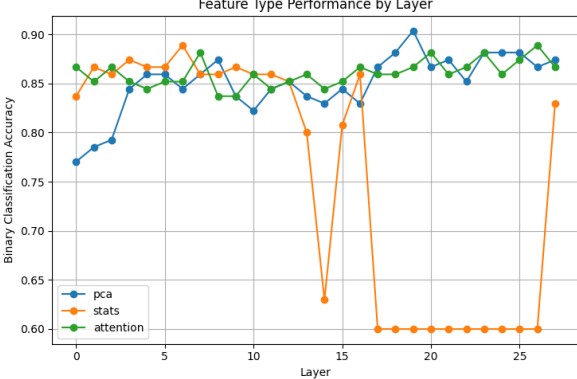

Figure 9: Layer-wise diagnostic results for **Fluency**. Evaluative signals show local peaks at Layer 17.

## J CASE ANALYSIS

This case compares three evaluators scoring the same model-generated response: a strong LLM judge (DeepSeek-V3), a probing-based judge that accesses intermediate representations of Qwen3-1.7B, and a prompt-based judge that relies on the same model's generated evaluation text. Consistent with our main hypothesis, the probing-based scores closely match the strong judge, indicating that the semantic and factual signals required for accurate evaluation are already encoded in the small model's latent states.

In contrast, the prompt-based evaluation from the same small model diverges: although its numeric scores remain plausible, the generated justifications introduce spurious concerns, overinterpret minor details, or provide unreliable reasoning. These deviations reflect the higher capacity demands of generating a coherent evaluation explanation. The pattern illustrates a core capacity asymmetry—evaluation depends on localized semantic cues accessible from intermediate representations, whereas producing well-formed meta-evaluative text requires long-range structuring and discourse planning that small models struggle to express reliably.

GSM8K EXAMPLE

**Question:**
Keaton climbed a 30 feet ladder twenty times while working at the construction site. Reece, also working at the same site, climbed a ladder 4 feet shorter than Keatoń's ladder 15 times. Whatś the total length of the ladders that both workers climbed in inches?

**Gold Answer:** 11880

**Model Prediction:**
Let's break it down step by step.

Keaton climbed a 30 feet ladder 20 times. To find the total length of the ladder he climbed, we need to multiply the height of the ladder by the number of times he climbed it:

30 feet/ladder × 20 times = 600 feet

Since there are 12 inches in 1 foot, we can convert the total length from feet to inches:

600 feet × 12 inches/foot = 7200 inches

Now, let's calculate the total length of the ladder Reece climbed. Reece climbed a ladder that was 4 feet shorter than Keaton's ladder, which means Reece's ladder was:

30 feet - 4 feet = 26 feet

Reece climbed the ladder 15 times, so the total length of the ladder he climbed is:

26 feet/ladder × 15 times = 390 feet

Converting this to inches:

390 feet × 12 inches/foot = 4680 inches

Now, let's add the total length of the ladders climbed by both workers:

7200 inches (Keaton) + 4680 inches (Reece) = 11,880 inches

So, the total length of the ladders that both workers climbed is 11,880 inches.

## DEEPSEEK-V3 OUTPUT

**Semantic Consistency (score 5):** Every step strictly follows the problem facts without inventing events, dropping givens, or adding unstated assumptions. The response correctly processes Keaton's 30-foot ladder climbed 20 times, calculates Reece's ladder as 4 feet shorter (26 feet) climbed 15 times, converts both totals from feet to inches using the standard conversion (12 inches per foot), and sums the results accurately. The final answer is fully grounded in the given problem.
**Logicality (score 5):** All inferences and arithmetic are valid; each step follows logically from prior steps. The response correctly interprets the problem, performs accurate calculations for both workers (including the conversion from feet to inches), and sums the results appropriately.
**Informativeness (score 5):** Full, verifiable step-by-step derivation showing all essential calculations: Keaton's total feet ($30 \times 20 = 600$), conversion to inches ($600 \times 12 = 7200$), Reece's ladder height ($30 - 4 = 26$), Reece's total feet ($26 \times 15 = 390$), conversion to inches ($390 \times 12 = 4680$), and final sum ($7200 + 4680 = 11880$). Anyone can re-check the answer from these clear, complete steps.
**Fluency (score 5):** Clear, grammatical, well-punctuated, and easy to follow; uses readable notation and step-by-step presentation that flows logically from one calculation to the next.
**Factuality (score 5):** All factual claims are correct and well-supported: accurate arithmetic ($30 \times 20 = 600$, $30 - 4 = 26$, $26 \times 15 = 390$, $600 \times 12 = 7200$, $390 \times 12 = 4680$, $7200 + 4680 = 11880$), correct unit conversion (12 inches per foot), and proper application of the problem's conditions. No errors, hallucinations, or unsupported assertions.

## QWEN3-1.7B PROBING EVALUATION

**Predicted scores:**

Semantic Consistency: 4
Logicality: 5
Informativeness: 5
Fluency: 5
Factuality: 5

QWEN3-1.7B PROMPT-BASED EVALUATION OUTPUT

> **Semantic Consistency (score 3):** Solution follows the problem facts. Minor consistency lapse: the rationale never states the tacit assumption that each 'climb' is a single ascent. Also, unit management is done per person instead of showing a single feet-to-inches conversion. These are small omissions that do not change the numeric result, so mid-level credit is appropriate.
>
> **Logicality (score 3):** Arithmetic steps are correct. Logical inefficiency exists: the chain converts each person to inches separately rather than adding feet then converting once — a harmless but unnecessary detour. Because all inferences are valid but the chain is slightly untidy, a middle score fits.
>
> **Informativeness (score 3):** Key calculations are present. However, the explanation omits one explicit grouped intermediate (explicitly writing 'total = 990 ft' before conversion is not shown in every version), and some steps are shown only implicitly. Enough detail to verify the answer, but not fully exhaustive — hence partial credit.
>
> **Fluency (score 4):** Language is clear, grammatical, and easy to follow. Sentences read naturally and numeric steps are separated. Minor repetition and slightly verbose conversion phrasing prevent a perfect score, but overall readability is high.
>
> **Factuality (score 2):** While the final numeric answer is correct, there are factual/imprecision concerns that lower confidence: (1) the rationale implicitly assumes 'climb' means one ascent and does not state this, which could be misread in other contexts; (2) converting per-person instead of converting total feet introduces an avoidable chance for unit-handling mistakes in variants of the problem. These represent multiple factual/imprecision issues significant enough to warrant a low factuality rating.

Figure: Complete evaluation outputs for the GSM8K case across three judges: DeepSeek-V3, Qwen3-1.7B (probing), and Qwen3-1.7B (prompting).

## K    PROMPT TEMPLATES

We provide the one-shot evaluation prompt templates for five aspects defined in Section 4.1 across three selected benchmarks: GSM8K, MATH, and GPQA. We use these prompts to get LLM evaluation results. Some parts of the prompt design are inspired by SOCREVAL(He et al., 2023).

It's noticeable that when we use these same prompts to prob small LMs, we found the prompts are too complex for small LMs to understand, which makes it difficult to acquire accurate probing results. Thus, for probing prompts, we remove the one-shot example and ask it to directly output the score, instead of JSON output. Specifically, for the probing evaluation prompt, we remove the "Example question", "Example generated response", and "Example representation". We also change the final output requirement to: "Now evaluate the Question and Generated response above based on the instruction. Return only the score."

**Evaluation Prompt of Semantic Consistency on GSM8K and MATH**

**Instruction:**
You are a judge that scores Semantic Consistency of a step-by-step rationale for a reasoning problem. Definition: Semantic Consistency = the solution steps and final answer must stay faithful to the problem facts (no inventing events, no dropping givens, no added unstated assumptions). A step is inconsistent if it contradicts the problem, introduces facts not present in the problem, or ignores givens. Scoring: integer 1–5.
5 — Every step and the final answer strictly follow the problem facts; no unstated assumptions or contradictions.
4 — Steps mostly follow the givens; one small unstated assumption that doesn't change the outcome.
3 — Minor omission or one mild contradiction that slightly weakens trust in the chain.
2 — Noticeable contradictions or added facts that affect the reasoning or outcome.
1 — Steps contradict the problem or introduce major unstated facts; answer not grounded in the problem.

**Example question:**
Natalia sold clips to 48 of her friends in April, and then she sold half as many clips in May. How many clips did Natalia sell altogether in April and May?

**Example generated response:**
She sold 48 in April. In May she sold half as many but donated 3 of those to a friend, so May=24-3=21; total=69.

**Example representation:**
{"score": 2, "justification": "Introduces an unstated donation (3) not in the problem; solution no longer strictly follows the given facts."}

**Question:** {Input question}

**Generated response:** {Input response}

Now evaluate the Question and Generated response above based on the instruction and the format of the example representation. Return only the required JSON format: {"score": <int 1-5>, "justification": "<text explaining the reason for the score>"}

Figure 10: Evaluation prompt of Semantic Consistency on GSM8K and MATH.

**Evaluation Prompt of Logicality on GSM8K and MATH**

**Instruction:**
You are a judge that scores Logicality of a step-by-step rationale for a reasoning problem. Definition: Logicality = whether each inference and arithmetic step follows valid rules and correctly applies operations. Penalize invalid deductions or misapplied reasoning. Scoring: integer 1–5.
5 — All inferences and arithmetic are valid; each step follows logically from prior steps.
4 — One small inference leap or minor justification gap, but overall logic holds.
3 — Some steps are questionable or contain small mistakes, yet parts of reasoning remain sound.
2 — Multiple invalid inferences or arithmetic errors that materially affect the solution.
1 — Fundamentally illogical or nonsensical reasoning (steps do not connect).

**Example question:**
Natalia sold clips to 48 of her friends in April, and then she sold half as many clips in May. How many clips did Natalia sell altogether in April and May?

**Example generated response:**
April = 48. 'Half as many in May' interpreted as add half of April to April, May = 48 + 48/2 = 72. Total = 48 + 72 = 120.

**Example representation:**
{"score": 1, "justification": "Misinterprets 'half as many' and performs an invalid reasoning step, producing an illogical total."}

**Question:** {Input question}

**Generated response:** {Input response}

Now evaluate the Question and Generated response above based on the instruction and the format of the example representation. Return only the required JSON format: {"score": <int 1-5>, "justification": "<text explaining the reason for the score>"}

Figure 11: Evaluation prompt of Logicality on GSM8K and MATH.

**Evaluation Prompt of Logicality on GPQA**

**Instruction:**
You are a judge that scores Logicality of a step-by-step rationale for a reasoning problem. Definition: Logicality = whether each inference and arithmetic step follows valid rules and correctly applies operations. Penalize invalid deductions or misapplied reasoning. Scoring: integer 1–5.
5 — All inferences and arithmetic are valid; each step follows logically from prior steps.
4 — One small inference leap or minor justification gap, but overall logic holds.
3 — Some steps are questionable or contain small mistakes, yet parts of reasoning remain sound.
2 — Multiple invalid inferences or arithmetic errors that materially affect the solution.
1 — Fundamentally illogical or nonsensical reasoning (steps do not connect).

**Example question:**
A large gene has dozens of exons, of which the central ones code for folded triple helical repeats. Which structure below is not involved in the proposed therapy?
Options:
polyA tail
R-loops
antisense
lariat

**Example generated response:**
Morpholino blocks exon, so lariat must be involved, polyA tail not used, R-loops skipped, antisense binds randomly.

**Example representation:**
{"score": 2, "justification": "Some reasoning steps are invalid or contradictory; e.g., 'antisense binds randomly' is unsupported and illogical in context."}

**Question**: {Input question}

**Generated response:** {Input response}

Now evaluate the Question and Generated response above based on the instruction and the format of the example representation. Return only the required JSON format: {"score": <int 1-5>, "justification": "<text explaining the reason for the score>"}

Figure 12: Evaluation prompt of Logicality on GPQA.

**Evaluation Prompt of Informativeness on GSM8K and MATH**

**Instruction:**
You are a judge that scores Informativeness of a step-by-step rationale for a reasoning problem. Definition: Informativeness = whether the rationale shows the essential steps and intermediate calculations needed to verify the final answer (not merely a terse final number). Reward verifiable, stepwise derivations. Scoring: integer 1–5.
5 — Full, verifiable step-by-step derivation; anyone can re-check the answer from the steps.
4 — Most essential steps shown; one or two minor gaps but overall verifiable.
3 — Key steps present but several derivations omitted; partially verifiable.
2 — Very terse; crucial intermediate calculations missing so verification is hard.
1 — Only an answer or irrelevant details; no usable derivation.

**Example question:**
Natalia sold clips to 48 of her friends in April, and then she sold half as many clips in May. How many clips did Natalia sell altogether in April and May?

**Example generated response:**
After thinking, the answer is 72.

**Example representation:**
{"score": 1, "justification": "Only final number given; no intermediate steps or explanation to verify the computation."}

**Question**: {Input question}

**Generated response:** {Input response}

Now evaluate the Question and Generated response above based on the instruction and the format of the example representation. Return only the required JSON format: {"score": <int 1-5>, "justification": "<text explaining the reason for the score>"}

Figure 13: Evaluation prompt of Informativeness on GSM8K and MATH.

**Evaluation Prompt of Informativeness on GPQA**

**Instruction:**
You are a judge that scores Informativeness of a step-by-step rationale for a reasoning problem. Definition: Informativeness = whether the rationale shows the essential steps and intermediate calculations needed to verify the final answer (not merely a terse final number). Reward verifiable, stepwise derivations. Scoring: integer 1–5.
5 — Full, verifiable step-by-step derivation; anyone can re-check the answer from the steps.
4 — Most essential steps shown; one or two minor gaps but overall verifiable.
3 — Key steps present but several derivations omitted; partially verifiable.
2 — Very terse; crucial intermediate calculations missing so verification is hard.
1 — Only an answer or irrelevant details; no usable derivation.

**Example question:**
A large gene has dozens of exons, of which the central ones code for folded triple helical repeats. Which structure below is not involved in the proposed therapy?
Options:
polyA tail
R-loops
antisense
lariat

**Example generated response:**
Answer: R-loops

**Example representation:**
{"score": 1, "justification": "Only the final answer is given; no step-by-step rationale to show how the answer was derived."}

**Question**: {Input question}

**Generated response:** {Input response}

Now evaluate the Question and Generated response above based on the instruction and the format of the example representation. Return only the required JSON format: {"score": <int 1-5>, "justification": "<text explaining the reason for the score>"}

Figure 14: Evaluation prompt of Informativeness on GPQA.

**Evaluation Prompt of Fluency on GSM8K and MATH**

**Instruction:**
You are a judge that scores Fluency (readability and clarity) of a step-by-step rationale for a reasoning problem. Definition: Fluency = the text is grammatical, clear, and easy to follow. Judge punctuation, sentence flow, readable notation and presentation. Fluency does NOT evaluate correctness. Scoring: integer 1–5.
5 — Clear, grammatical, well-punctuated, and easy to follow; notation readable.
4 — Mostly clear with small phrasing or punctuation issues.
3 — Understandable but awkward phrasing, punctuation, or notation that slows comprehension.
2 — Hard to follow; many grammatical issues or poor notation.
1 — Unreadable or incoherent language.

**Example question:**
Natalia sold clips to 48 of her friends in April, and then she sold half as many clips in May. How many clips did Natalia sell altogether in April and May?

**Example generated response:**
sold 48 april then half may 24 total 72

**Example representation:**
{"score": 2, "justification": "Ungrammatical and terse; the meaning is recoverable but phrasing and punctuation are poor."}

**Question**: {Input question}

**Generated response:** {Input response}

Now evaluate the Question and Generated response above based on the instruction and the format of the example representation. Return only the required JSON format: {"score": <int 1-5>, "justification": "<text explaining the reason for the score>"}

Figure 15: Evaluation prompt of Fluency on GSM8K and MATH.

**Evaluation Prompt of Fluency on GPQA**

**Instruction:**
You are a judge that scores Fluency (readability and clarity) of a step-by-step rationale for a reasoning problem. Definition: Fluency = the text is grammatical, clear, and easy to follow. Judge punctuation, sentence flow, readable notation and presentation. Fluency does NOT evaluate correctness. Scoring: integer 1–5.
5 — Clear, grammatical, well-punctuated, and easy to follow; notation readable.
4 — Mostly clear with small phrasing or punctuation issues.
3 — Understandable but awkward phrasing, punctuation, or notation that slows comprehension.
2 — Hard to follow; many grammatical issues or poor notation.
1 — Unreadable or incoherent language.

**Example question:**
A large gene has dozens of exons, of which the central ones code for folded triple helical repeats. Which structure below is not involved in the proposed therapy?
Options:
polyA tail
R-loops
antisense
lariat

**Example generated response:**
use morpholino. not involve r-loops

**Example representation:**
{"score": 2, "justification": "Short, ungrammatical phrasing and poor punctuation make it hard to read, although meaning is recoverable."}

**Question:** {Input question}

**Generated response:** {Input response}

Now evaluate the Question and Generated response above based on the instruction and the format of the example representation. Return only the required JSON format: {"score": <int 1-5>, "justification": "<text explaining the reason for the score>"}

Figure 16: Evaluation prompt of Fluency on GPQA.

**Evaluation Prompt of Factuality on GSM8K and MATH**

**Instruction:**
You are a judge that scores Factuality of a step-by-step rationale for a reasoning problem. Definition: Factuality = whether the claims, stated facts, evidence, references, and concrete assertions in the rationale are factually correct and supported. Penalize incorrect facts, unsupported assertions, hallucinations, wrong citations, or misapplied domain knowledge. Scoring: integer 1–5.
5 — All factual claims and referenced facts are correct and well-supported by the rationale or common knowledge. No hallucinations.
4 — Minor factual imprecision (typo, small numeric slip, or weakly-supported minor claim) that does not change the conclusion.
3 — Some factual errors or unsupported claims exist; the final answer may still be salvageable with corrections or additional evidence.
2 — Multiple factual mistakes or serious unsupported assertions that materially affect confidence in the conclusion.
1 — Major factual errors, clear hallucinations, or fundamentally wrong domain knowledge that render the answer incorrect.

**Example question:**
Natalia sold clips to 48 of her friends in April, and then she sold half as many clips in May. How many clips did Natalia sell altogether in April and May?

**Example generated response:**
She sold 48/2 = 20 clips in May. Natalia sold 48+20 = 68 clips altogether. #### 68

**Example representation:**
{"score": 1, "justification": "Arithmetic is incorrect (48/2 $\neq$ 20), so factual claims and final numeric result are wrong."}

**Question:** {Input question}

**Generated response:** {Input response}

Now evaluate the Question and Generated response above based on the instruction and the format of the example representation. Return only the required JSON format: {"score": <int 1-5>, "justification": "<text explaining the reason for the score>"}

Figure 17: Evaluation prompt of Factuality on GSM8K and MATH.

**Evaluation Prompt of Factuality on GPQA**

**Instruction:**
You are a judge that scores Factuality of a step-by-step rationale for a reasoning problem. Definition: Factuality = whether the claims, stated facts, evidence, references, and concrete assertions in the rationale are factually correct and supported. Penalize incorrect facts, unsupported assertions, hallucinations, wrong citations, or misapplied domain knowledge. Scoring: integer 1–5.
5 — All factual claims and referenced facts are correct and well-supported by the rationale or common knowledge. No hallucinations.
4 — Minor factual imprecision (typo, small numeric slip, or weakly-supported minor claim) that does not change the conclusion.
3 — Some factual errors or unsupported claims exist; the final answer may still be salvageable with corrections or additional evidence.
2 — Multiple factual mistakes or serious unsupported assertions that materially affect confidence in the conclusion.
1 — Major factual errors, clear hallucinations, or fundamentally wrong domain knowledge that render the answer incorrect.

**Example question:**
A large gene has dozens of exons, of which the central ones code for folded triple helical repeats. Which structure below is not involved in the proposed therapy?
Options:
polyA tail
R-loops
antisense
lariat

**Example generated response:**
The polyA tail is not used; antisense and R-loops both irrelevant; lariat forms irrelevant loops.

**Example representation:**
{"score": 3, "justification": "Some claims are factually imprecise (e.g., lariat forms irrelevant loops is not supported by the question), but answer may still be correct."}

**Question**: {Input question}

**Generated response:** {Input response}

Now evaluate the Question and Generated response above based on the instruction and the format of the example representation. Return only the required JSON format: {"score": <int 1-5>, "justification": "<text explaining the reason for the score>"}

Figure 18: Evaluation prompt of Factuality on GPQA.

