# OpenReview forum: "Rethinking LLM-as-a-Judge: Representation-as-a-Judge with Small Language Models via Semantic Capacity Asymmetry"
_ICLR.cc/2026/Conference — ICLR 2026 Poster_

### Official Review · Reviewer_DGMU · 2025-10-24

**Soundness:** 3
**Presentation:** 3
**Contribution:** 3
**Rating:** 6
**Confidence:** 5

**Summary:**

This paper challenges the conventional LLM-as-a-Judge paradigm by showing that small language models (SLMs), despite weak generative ability, encode rich evaluative signals in their hidden representations. It proposes the Semantic Capacity Asymmetry Hypothesis, suggesting that evaluation requires less semantic capacity than generation. The authors introduce INSPECTOR, a probing-based framework that predicts aspect-level evaluation scores using internal representations of SLMs. Experiments on reasoning datasets (GSM8K, MATH, GPQA) demonstrate that INSPECTOR approaches the performance of large LLM judges while being more efficient and interpretable. This work advances a Representation-as-a-Judge paradigm that enables cost-effective, scalable, and transparent evaluation of model outputs.

**Strengths:**

1. The paper introduces the Representation-as-a-Judge framework which is a new way of performing evaluation without decoding, shifting the paradigm from generation-based judgment to representation-based probing.
2. The paper demonstrates consistent and large gains over prompt-based baselines across multiple reasoning datasets, showing that evaluative signals exist in intermediate representations.
3. The proposed approach significantly reduces inference cost, avoids dependence on proprietary LLMs (like GPT-4), and provides interpretable layer-level insights.
4. Uses binary probing classifiers to filter low-quality data effectively, improving downstream supervised fine-tuning.

**Weaknesses:**

1. Despite promoting efficiency, the pipeline still depends on large LLMs (like DeepSeek-V3) to provide "gold" evaluation scores. Gathering these scores is itself expensive.
2. Experiments focus only on mathematical reasoning (GSM8K, MATH, GPQA); it’s unclear whether the approach generalizes to open-domain, creative, dialogue tasks or safety evaluation.
3. While simple probes (logistic regression) are interpretable, they may underfit complex evaluative signals, limiting the upper bound of achievable fidelity.
4. The paper lacks qualitative analysis and error analysis. Observing some samples where these models perform much better than baselines, or looking at error buckets would be nice.

**Questions:**

1. Could multi-task or cross-aspect probing improve performance compared to aspect-specific classifiers?
2. How stable are the layer rankings across different datasets — do the same layers consistently encode evaluative information?
3. Can this method be extended to explain why a response was scored poorly (e.g., highlighting reasoning errors)?
4. Could the representation-based judge itself be used as a reward model in reinforcement learning from human feedback (RLHF)?

---

> ### Author Response · Authors · 2025-11-21
> **Response 1 to Reviewer Comments**
>
> We appreciate the reviewer’s positive assessment of the Representation-as-a-Judge paradigm and the strengths highlighted regarding efficiency, interpretability, and empirical performance. The feedback also raises several valuable points about the reliance on LLM-provided supervision, the behavior of fine-grained probes, the design choices within INSPECTOR, and the scope of generalization beyond reasoning tasks. We address each of these concerns in detail below and outline the clarifications and revisions that will strengthen the final version of the paper.
>
> **Weaknesses:**
>
> **1.** Despite promoting efficiency, the pipeline still depends on large LLMs (like DeepSeek-V3) to provide "gold" evaluation scores. Gathering these scores is itself expensive.
>
> **Response:**
> Thank you for pointing this out. Our approach does require a strong LLM to generate the initial evaluation scores, and we agree that obtaining these labels has a cost. However, this dependency occurs only once during data creation. After the initial labels are produced, INSPECTOR enables repeated evaluation using small, open-source models, eliminating the need to query a large LLM for every new sample. Moreover, DeepSeek-V3’s low token price makes full-dataset evaluation affordable—collecting all scores for GSM8K and MATH costs under $15, respectively.
>
> **2.** Experiments focus only on mathematical reasoning (GSM8K, MATH, GPQA); it’s unclear whether the approach generalizes to open-domain, creative, dialogue tasks or safety evaluation.
>
> **Response:**
> Thank you for raising this point. To provide initial evidence of generalization, we additionally evaluated our method on the open-ended, reference-free generation benchmark AlpacaEval 2.0 using the same experimental settings as in our reasoning experiments. The results are presented in the tables below.
>
> **Multiclass Classification (score=1-5)** on AlpacaEval 2.0:
> | | Semantic Consistency | Logicality | Informativeness | Fluency | Factuality |
> |:---:|:---:|:---:|:---:|:---:|:---:|
> | RoBERTa | 7.56% | 8.08% | 8.32% | 11.69% | 8.08% |
> | Qwen3-0.6B (Tuning) | 14.29% | 11.11% | 9.93% | 10.61% | 12.70% |
> | Qwen3-0.6B (Prompt) | 22.40% | 11.11% | 16.94% | 11.82% | 8.89% |
> | Qwen3-0.6B (Probing) | 36.84% | 42.59% | 46.71% | 61.21% | 23.33% |
> | Llama-3.2-1B-Instruct (Prompt) | 12.50% | 13.33% | 8.37% | 16.97% | 0.20% |
> | Llama-3.2-1B-Instruct (Probing) | 35.71% | 55.56% | 46.08% | 61.21% | 28.57% |
> | Qwen3-1.7B (Prompt) | 7.14% | 20.11% | 22.70% | 25.19% | 20.00% |
> | Qwen3-1.7B (Probing) | 35.12% | 45.93% | 42.01% | 65.10% | 23.70% |
> | Llama-3.1-8B-Instruct (Prompt) | 14.92% | 17.99% | 18.34% | 23.38% | 14.29% |
> | Llama-3.1-8B-Instruct (Probing) | 34.69% | 53.70% | 45.90% | 57.32% | 22.96% |
>
> **Binary-Classification (high vs low quality)** on AlpacaEval 2.0:
> | | Semantic Consistency | Logicality | Informativeness | Fluency | Factuality |
> |:---:|:---:|:---:|:---:|:---:|:---:|
> | RoBERTa | 41.56% | 39.68% | 46.58% | 49.49% | 39.68% |
> | Qwen3-0.6B (Tuning) | 35.24% | 43.00% | 45.41% | 28.48% | 27.78% |
> | Qwen3-0.6B (Prompt) | 55.24% | 43.00% | 49.93% | 46.36% | 44.44% |
> | Qwen3-0.6B (Probing) | 61.42% | 55.56% | 62.76% | 91.06% | 55.56% |
> | Llama-3.2-1B-Instruct (Prompt) | 49.20% | 34.19% | 59.26% | 64.24% | 39.68% |
> | Llama-3.2-1B-Instruct (Probing) | 67.14% | 70.42% | 67.44% | 81.82% | 65.80% |
> | Qwen3-1.7B (Prompt) | 50.24% | 65.80% | 45.03% | 45.45% | 55.56% |
> | Qwen3-1.7B (Probing) | 67.14% | 75.93% | 70.71% | 91.06% | 77.78% |
> | Llama-3.1-8B-Instruct (Prompt) | 39.77% | 48.15% | 58.34% | 62.42% | 27.35% |
> | Llama-3.1-8B-Instruct (Probing) | 70.16% | 77.78% | 69.96% | 82.12% | 64.07% |
>
> The results demonstrate that our probing methods yield consistent improvements over other baselines and exhibit trends similar to those reported in our paper. This further supports the effectiveness and robustness of our approach on a broader range of domains and tasks, including reference-free open-ended benchmarks.
>
> **3.** While simple probes (logistic regression) are interpretable, they may underfit complex evaluative signals, limiting the upper bound of achievable fidelity.
>
> **Response:**
> Thank you for the observation. Logistic regression is indeed a simple probe, and it may not capture all of the evaluative structure present in the representations. In this work, we intentionally adopt lightweight and interpretable probes to examine whether the relevant signals are already linearly accessible in small-model hidden states. The results indicate that a substantial portion of the signal can be recovered with such simple models. More expressive probe architectures, such as decomposition-based or variational approaches (e.g., VAE-style probes), could capture additional structure and potentially improve fidelity. Exploring these richer probes is a natural direction for future work, and we will clarify this point in the revision.

---

> ### Author Response · Authors · 2025-11-21
> **Response 2 to Reviewer Comments**
>
> **Weaknesses:**
>
> **4.** The paper lacks qualitative analysis and error analysis. Observing some samples where these models perform much better than baselines, or looking at error buckets would be nice.
>
> **Response:**
> Thank you for the suggestion. We agree that qualitative examples and a brief error analysis would help illustrate how the representation-based judge behaves in practice. Due to space constraints in the rebuttal, we cannot include them here, but in the revision, we will add several representative cases showing where INSPECTOR outperforms prompting baselines, together with a small number of illustrative error cases. We believe these additions will make the behavior of the method clearer without changing the overall conclusions.
>
> **Questions:**
>
> **1.** Could multi-task or cross-aspect probing improve performance compared to aspect-specific classifiers?
>
> **Response:**
> Thank you for the question. In INSPECTOR, each evaluation aspect is conditioned on its own scoring prompt, which produces aspect-specific hidden representations from the frozen LLM. Because different prompts activate different parts of the representation space, the probing tasks for different aspects are not naturally aligned. For this reason, we train separate classifiers so that each dimension can be modeled from the representations specifically induced for it, keeping the evaluative signals disentangled and interpretable.
>
> Multi-task or cross-aspect probing would require a shared representation space across aspects, which is not the case under prompt-conditioned evaluation. As such, aspect-specific probes are a more appropriate choice in our setting. We will clarify this design rationale in the revision.
>
> **2.** How stable are the layer rankings across different datasets — do the same layers consistently encode evaluative information?
>
> **Response:**
> We appreciate the reviewer’s question. Based on our observations of LLM probing across multiple datasets, we find that mid-layer representations often contain stronger evaluative signals correlated with the final gold scores, compared to early or late layers. However, the specific layers that perform best are not strictly invariant across datasets. For example, for Qwen3-1.7B on MATH, the top layers for semantic consistency include layers: 15, 17, 16, whereas on GSM8K the top layers include: 20, 15,16. This suggests that while mid layers generally encode rich evaluative information, the exact optimal layers may vary depending on datasets.
>
> We have already included probing results for MATH in Section 6.1 and in Appendix H of our submission. In the revised version, we will add additional probing analyses for the other datasets to further support these findings.
>
> **3.** Can this method be extended to explain why a response was scored poorly (e.g., highlighting reasoning errors)?
>
> **Response:**
> Thank you for the question. INSPECTOR, in its current form, predicts aspect-level scores, but it does not aim to localize or characterize specific reasoning errors. At the same time, the probing results suggest that different types of failures may manifest as distinguishable patterns in the representation space. Extending the approach toward explanation, such as analyzing whether particular regions of the hidden states, attention patterns, or low-dimensional projections correspond to systematic error types, would require additional mechanisms for attributing internal signals back to segments of the model output. Exploring whether representation geometry can reveal reasoning-error categories is a promising direction for future work, and we will clarify this distinction in the revision.
>
> **4.** Could the representation-based judge itself be used as a reward model in reinforcement learning from human feedback (RLHF)?
>
> **Response:**
> Thank you for raising this question. The outputs of INSPECTOR provide useful signals about response quality, but they are not directly suitable as an RLHF reward model. In particular, the five-level predictions do not achieve sufficient accuracy to be used as stable scalar rewards, and the binary classifiers often produce sharp probability outputs that are not calibrated for policy optimization. A more appropriate way to leverage INSPECTOR in the RLHF setting would be to use the binary probe as an automatic preference annotator. The binary predictions provide a relative preference signal that can be converted into pairwise labels, which can then be used to train a dedicated reward model with standard preference-based objectives. While this extension is outside the scope of the present work, it represents a feasible direction for future research.

---

### Official Review · Reviewer_DiLJ · 2025-10-28

**Soundness:** 3
**Presentation:** 3
**Contribution:** 2
**Rating:** 6
**Confidence:** 3

**Summary:**

This paper proposes a new evaluation paradigm called Representation-as-a-Judge, which shifts from relying on large language model (LLM) generation to using internal representations from small LMs (SLMs) for evaluation. The authors introduce the Semantic Capacity Asymmetry Hypothesis, arguing that evaluation requires less semantic capacity than generation. They instantiate this idea in INSPECTOR, a probing-based framework that trains lightweight classifiers on SLM hidden states to predict quality scores (e.g., logicality, factuality) derived from a strong LLM judge. Experiments on GSM8K, MATH, and GPQA show that SLM probes outperform prompting baselines and closely approximate full LLM evaluations, especially in binary classification tasks. The approach also proves useful for filtering training data to improve downstream supervised fine-tuning.

**Strengths:**

1. The Representation-as-a-Judge framework is technically sound, which could be an alternative to the prevalent “LLM-as-a-Judge” approach.

2. The approach enables efficient evaluation using smaller, open-source models instead of proprietary LLMs.

3. The empirical results are strong, showing significant gains over some prompting and fine-tuning baselines.

**Weaknesses:**

1. A limitation is that the method still requires a powerful LLM in the loop to obtain initial evaluation scores (for training data). This paper assumes the LLM’s scores are gold-standard. It would strengthen the work to either validate against human ratings or discuss the implications of this dependency.

2. The probing classifiers achieve relatively low accuracy on fine-grained multiclass (1–5) predictions. This might limit the method’s use if one requires precise scoring, and also indicates some inherent issue that the approach doesn’t fully match the large model in terms of detailed gradations of quality.

3. The INSPECTOR pipeline introduces additional complexity and tuning that a direct LLM judge does not require. One must decide how to pool representations, which layers to select, what classifier to use, etc. This paper doesn’t deeply explore the sensitivity to these choices.

**Questions:**

1. Have the authors evaluated or considered how the probe’s judgments (or the large LLM’s scores it learns from) align with human evaluations of the responses?

2. The experiments focus on mathematical reasoning problems. How well do the authors expect the Representation-as-a-Judge approach to transfer to other domains or tasks?

3. Did the authors experiment with regression instead of classification to predict aspect scores on a continuous scale?

---

> ### Author Response · Authors · 2025-11-21
> **Response 1 to Reviewer Comments**
>
> We appreciate the reviewer’s positive assessment of the technical soundness and empirical strength of the approach, as well as the constructive suggestions for improvement. We address each of these points in detail below.
>
> **Weaknesses:**
>
> **1.** A limitation is that the method still requires a powerful LLM in the loop to obtain initial evaluation scores (for training data). This paper assumes the LLM’s scores are gold-standard. It would strengthen the work to either validate against human ratings or discuss the implications of this dependency.
>
> **Response:**
> Thank you for pointing out this limitation. Our method requires a strong LLM to provide the initial evaluation scores, and we agree that this dependency is important to acknowledge. In practice, many recent evaluation frameworks rely on high-capability LLM judges as a cost-effective proxy for human ratings, since their outputs have been shown to correlate well with human preferences. In this sense, using LLM-based labels is consistent with common practice rather than introducing an additional assumption specific to our method.
>
> Our objective is to examine whether the evaluations produced by a strong judge can be recovered from small-model representations, rather than to establish a new notion of ground-truth quality. Within this scope, LLM-provided labels serve as a practical source of supervision. We will clarify this perspective and discuss the implications of this dependency in the revision.
>
> **2.** The probing classifiers achieve relatively low accuracy on fine-grained multiclass (1–5) predictions. This might limit the method’s use if one requires precise scoring, and also indicates some inherent issue that the approach doesn’t fully match the large model in terms of detailed gradations of quality.
>
> **Response:**
> Thank you for highlighting this point. Fine-grained 1–5 scoring is a challenging setting, and our probing classifiers show lower accuracy there. Prior work has noted that absolute quality scores tend to be more variable than pairwise preference judgments, even for strong LLM judges. For example, Zheng et al. [1] report high human–LLM agreement in pairwise comparison settings, whereas finer-grained absolute scoring is generally observed to be less stable. Our goal is therefore not to fully reproduce the entire 1–5 distribution, but to assess whether the semantic signals needed for reliable judging can be accessed through small-model representations.
>
> Despite the difficulty of fine-grained scoring, INSPECTOR achieves strong consistency with the large LLM on binary and coarse-grained evaluation, which are widely used in practical LLM assessment. In applied scenarios, these properties make representation-based judges particularly suitable for large-scale filtering: a small model can efficiently perform initial screening, after which a strong evaluator may be used for finer-grained assessment when needed. We will clarify this intended usage and the limitations of the 1–5 setting in the revision.
>
> **[1]** L. Zheng et al. _Judging LLM-as-a-Judge with MT-Bench and Chatbot Arena._ NeurIPS, 2023.
>
> **3.** The INSPECTOR pipeline introduces additional complexity and tuning that a direct LLM judge does not require. One must decide how to pool representations, which layers to select, what classifier to use, etc. This paper doesn’t deeply explore the sensitivity to these choices.
>
> **Response:**
> Thank you for pointing this out. INSPECTOR involves choices such as pooling strategy, layer selection, and classifier type, and different configurations do lead to some variation, as shown in Figure 4. However, the cost of exploring these options is extremely low in practice. The hidden representations only need to be computed once on GPUs, and then try different pooling schemes and lightweight classifiers on a single CPU. The average time to train a probing classifier on one aspect is 10.86 seconds based on the scikit-learn tools. We will clarify this low-overhead point in the revision.
>
> **Questions:**
>
> **1.** Have the authors evaluated or considered how the probe’s judgments (or the large LLM’s scores it learns from) align with human evaluations of the responses?
>
> **Response:**
> Thank you for raising this point. We have considered how our evaluations relate to human judgments. Prior work has shown that high-capability LLM judges often exhibit strong agreement with human preferences, and using LLM-based judgments as a proxy for human annotation has become a standard and cost-effective practice in recent evaluation frameworks. Since our goal is to investigate whether a small model can recover the judgments of a strong LLM evaluator, rather than to replicate human annotation itself, we adopt DeepSeek-V3 as the supervising judge and do not conduct additional human labeling.
>
> If the reviewer considers a limited human-annotation comparison necessary, we are happy to include a small-scale human study during the rebuttal period.

---

> ### Author Response · Authors · 2025-11-21
> **Response 2 to Reviewer Comments**
>
> **Questions:**
>
> **2.** The experiments focus on mathematical reasoning problems. How well do the authors expect the Representation-as-a-Judge approach to transfer to other domains or tasks?
>
> **Response:**
> Thank you for the question. Although our primary experiments focus on mathematical reasoning, the Representation-as-a-Judge approach is not tied to this particular domain. To provide initial evidence, we also evaluated our approach on the open-ended generation benchmark AlpacaEval 2.0 using the same setup as in our reasoning experiments. The results are shown in the tables below.
>
> **Multiclass Classification (score=1-5)** on AlpacaEval 2.0:
> | | Semantic Consistency | Logicality | Informativeness | Fluency | Factuality |
> |:---:|:---:|:---:|:---:|:---:|:---:|
> | RoBERTa | 7.56% | 8.08% | 8.32% | 11.69% | 8.08% |
> | Qwen3-0.6B (Tuning) | 14.29% | 11.11% | 9.93% | 10.61% | 12.70% |
> | Qwen3-0.6B (Prompt) | 22.40% | 11.11% | 16.94% | 11.82% | 8.89% |
> | Qwen3-0.6B (Probing) | 36.84% | 42.59% | 46.71% | 61.21% | 23.33% |
> | Llama-3.2-1B-Instruct (Prompt) | 12.50% | 13.33% | 8.37% | 16.97% | 0.20% |
> | Llama-3.2-1B-Instruct (Probing) | 35.71% | 55.56% | 46.08% | 61.21% | 28.57% |
> | Qwen3-1.7B (Prompt) | 7.14% | 20.11% | 22.70% | 25.19% | 20.00% |
> | Qwen3-1.7B (Probing) | 35.12% | 45.93% | 42.01% | 65.10% | 23.70% |
> | Llama-3.1-8B-Instruct (Prompt) | 14.92% | 17.99% | 18.34% | 23.38% | 14.29% |
> | Llama-3.1-8B-Instruct (Probing) | 34.69% | 53.70% | 45.90% | 57.32% | 22.96% |
>
> **Binary-Classification (high vs low quality)** on AlpacaEval 2.0:
> | | Semantic Consistency | Logicality | Informativeness | Fluency | Factuality |
> |:---:|:---:|:---:|:---:|:---:|:---:|
> | RoBERTa | 41.56% | 39.68% | 46.58% | 49.49% | 39.68% |
> | Qwen3-0.6B (Tuning) | 35.24% | 43.00% | 45.41% | 28.48% | 27.78% |
> | Qwen3-0.6B (Prompt) | 55.24% | 43.00% | 49.93% | 46.36% | 44.44% |
> | Qwen3-0.6B (Probing) | 61.42% | 55.56% | 62.76% | 91.06% | 55.56% |
> | Llama-3.2-1B-Instruct (Prompt) | 49.20% | 34.19% | 59.26% | 64.24% | 39.68% |
> | Llama-3.2-1B-Instruct (Probing) | 67.14% | 70.42% | 67.44% | 81.82% | 65.80% |
> | Qwen3-1.7B (Prompt) | 50.24% | 65.80% | 45.03% | 45.45% | 55.56% |
> | Qwen3-1.7B (Probing) | 67.14% | 75.93% | 70.71% | 91.06% | 77.78% |
> | Llama-3.1-8B-Instruct (Prompt) | 39.77% | 48.15% | 58.34% | 62.42% | 27.35% |
> | Llama-3.1-8B-Instruct (Probing) | 70.16% | 77.78% | 69.96% | 82.12% | 64.07% |
>
> The results demonstrate that our probing methods yield consistent improvements over other baselines and exhibit trends similar to those reported in our paper. This further supports the effectiveness and robustness of our approach on a broader range of domains and tasks, including reference-free open-ended benchmarks.
>
> **3.** Did the authors experiment with regression instead of classification to predict aspect scores on a continuous scale?
>
> **Response:**
> Thank you for highlighting this possible direction. To address your suggestion, we converted our classification setup into a regression setup and conducted experiments on the GSM8K benchmark using Qwen3-1.7B and Llama-3.2-1B-Instruct. In this setting, we replaced the cross-entropy loss with mean squared error (MSE) for the probing classifiers. The results are shown in the following table:
>
> **Multiclass Classification (score=1-5)** on GSM8K:
> | | Semantic Consistency | Logicality | Informativeness | Fluency | Factuality |
> |:---:|:---:|:---:|:---:|:---:|:---:|
> | Llama-3.2-1B-Instruct (Prompt) | 6.33% | 9.55% | 23.54% | 17.11% | 11.76% |
> | Llama-3.2-1B-Instruct (Classification Probing) | 38.04% | 47.16% | 42.33% | 53.29% | 43.29% |
> | Llama-3.2-1B-Instruct (**Regression Probing**) | 26.02% | 46.15% | 38.90% | 41.18% | 36.47% |
> | Qwen3-1.7B (Prompt) | 16.18% | 25.00% | 13.88% | 22.27% | 30.46% |
> | Qwen3-1.7B (Classification Probing) | 42.98% | 49.86% | 46.51% | 47.39% | 48.86% |
> | Qwen3-1.7B (**Regression Probing**) | 29.64% | 46.15% | 45.31% | 41.18% | 37.65% |
>
> **Binary-Classification (high vs low quality)** on GSM8K:
> | | Semantic Consistency | Logicality | Informativeness | Fluency | Factuality |
> |:---:|:---:|:---:|:---:|:---:|:---:|
> | Llama-3.2-1B-Instruct (Prompt) | 43.57% | 43.08% | 42.79% | 37.65% | 44.44% |
> | Llama-3.2-1B-Instruct (Classification Probing) | 82.35% | 65.00% | 72.08% | 82.35% | 71.42% |
> | Llama-3.2-1B-Instruct (**Regression Probing**) | 73.74% | 50.67% | 69.91% | 76.63% | 63.90% |
> | Qwen3-1.7B (Prompt) | 38.95% | 61.54% | 38.94% | 41.58% | 69.85% |
> | Qwen3-1.7B (Classification Probing) | 76.13% | 72.78% | 72.08% | 88.32% | 81.26% |
> | Qwen3-1.7B (**Regression Probing**) | 67.90% | 65.85% | 67.74% | 76.63% | 70.88% |
>
> The results indicate that because our aspect scores are inherently discrete (1–5), regression performs slightly worse than our original classification formulation. Nevertheless, the regression models still outperform the baselines, further demonstrating the robustness of our probing approach across alternative prediction paradigms.

---

### Official Review · Reviewer_CaW1 · 2025-10-30

**Soundness:** 3
**Presentation:** 3
**Contribution:** 3
**Rating:** 6
**Confidence:** 4

**Summary:**

This paper proposes INSPECTOR, a probing-based framework that evaluates model outputs using internal representations instead of text generation. It introduces the Semantic Capacity Asymmetry Hypothesis, suggesting that evaluation requires less semantic capacity than generation. By leveraging small open-source LMs, INSPECTOR enables lightweight, interpretable, and scalable evaluation, achieving performance comparable to large proprietary models.

**Strengths:**

1. The paper reframes model evaluation as Representation-as-a-Judge, shifting from prompt-based evaluation to representation-based probing.
2. By leveraging internal representations from small open-source LMs instead of large proprietary models, INSPECTOR offers a lightweight and interpretable alternative that significantly reduces computational cost.
3. The framework achieves high predictive accuracy on reasoning benchmarks (e.g., GSM8K, MATH, GPQA), demonstrating that small models (1.7B) can approximate large-scale evaluators.
4. The proposed method can enhance LLM evaluation pipelines, reduce dependence on expensive closed models, and support data filtering for downstream tasks.

**Weaknesses:**

1. Experiments are primarily focused on reasoning benchmarks; it is unclear whether the method generalizes to other domains such as dialogue, summarization, or open-ended generation.
2. The performance quality of the judge (small LM) could be dependent on how the evaluation criteria are established.

**Questions:**

1. Is there a specific reason why the paper focuses only on decoder-only models instead of considering encoder-only architectures?
2. I believe the Semantic Capacity Asymmetry Hypothesis aligns with a well-established understanding in LLM research. It would be helpful to cite prior work that supports this perspective.

---

> ### Author Response · Authors · 2025-11-21
> **Response 1 to Reviewer Comments**
>
> We appreciate the positive remarks on our framework, its practical efficiency benefits, and the empirical strength of our probing approach. The reviewer’s comments also highlight several important points regarding generalization, dependence on evaluation criteria, architectural considerations, and theoretical grounding. We address each item in detail below and outline the revisions we will incorporate to improve clarity and completeness further.
>
> **Weaknesses:**
>
> **1.** Experiments are primarily focused on reasoning benchmarks; it is unclear whether the method generalizes to other domains such as dialogue, summarization, or open-ended generation.
>
> **Response:**
> Thank you for the helpful suggestions regarding evaluating our method on broader and more general tasks. While our primary focus is on reasoning evaluation, we additionally evaluated our approach on the open-ended generation benchmark AlpacaEval 2.0, using the same experimental settings as the three reasoning benchmarks reported in the paper. The results are presented in the tables below.
>
> **Multiclass Classification (score=1-5)** on AlpacaEval 2.0:
> | | Semantic Consistency | Logicality | Informativeness | Fluency | Factuality |
> |:---:|:---:|:---:|:---:|:---:|:---:|
> | RoBERTa | 7.56% | 8.08% | 8.32% | 11.69% | 8.08% |
> | Qwen3-0.6B (Tuning) | 14.29% | 11.11% | 9.93% | 10.61% | 12.70% |
> | Qwen3-0.6B (Prompt) | 22.40% | 11.11% | 16.94% | 11.82% | 8.89% |
> | Qwen3-0.6B (Probing) | 36.84% | 42.59% | 46.71% | 61.21% | 23.33% |
> | Llama-3.2-1B-Instruct (Prompt) | 12.50% | 13.33% | 8.37% | 16.97% | 0.20% |
> | Llama-3.2-1B-Instruct (Probing) | 35.71% | 55.56% | 46.08% | 61.21% | 28.57% |
> | Qwen3-1.7B (Prompt) | 7.14% | 20.11% | 22.70% | 25.19% | 20.00% |
> | Qwen3-1.7B (Probing) | 35.12% | 45.93% | 42.01% | 65.10% | 23.70% |
> | Llama-3.1-8B-Instruct (Prompt) | 14.92% | 17.99% | 18.34% | 23.38% | 14.29% |
> | Llama-3.1-8B-Instruct (Probing) | 34.69% | 53.70% | 45.90% | 57.32% | 22.96% |
>
> **Binary-Classification (high vs low quality)** on AlpacaEval 2.0:
> | | Semantic Consistency | Logicality | Informativeness | Fluency | Factuality |
> |:---:|:---:|:---:|:---:|:---:|:---:|
> | RoBERTa | 41.56% | 39.68% | 46.58% | 49.49% | 39.68% |
> | Qwen3-0.6B (Tuning) | 35.24% | 43.00% | 45.41% | 28.48% | 27.78% |
> | Qwen3-0.6B (Prompt) | 55.24% | 43.00% | 49.93% | 46.36% | 44.44% |
> | Qwen3-0.6B (Probing) | 61.42% | 55.56% | 62.76% | 91.06% | 55.56% |
> | Llama-3.2-1B-Instruct (Prompt) | 49.20% | 34.19% | 59.26% | 64.24% | 39.68% |
> | Llama-3.2-1B-Instruct (Probing) | 67.14% | 70.42% | 67.44% | 81.82% | 65.80% |
> | Qwen3-1.7B (Prompt) | 50.24% | 65.80% | 45.03% | 45.45% | 55.56% |
> | Qwen3-1.7B (Probing) | 67.14% | 75.93% | 70.71% | 91.06% | 77.78% |
> | Llama-3.1-8B-Instruct (Prompt) | 39.77% | 48.15% | 58.34% | 62.42% | 27.35% |
> | Llama-3.1-8B-Instruct (Probing) | 70.16% | 77.78% | 69.96% | 82.12% | 64.07% |
>
> The results demonstrate that our probing methods yield consistent improvements over other baselines and exhibit trends similar to those reported in our paper. This further supports the effectiveness and robustness of our approach on a broader range of domains and tasks, including reference-free open-ended benchmarks.
>
> **2.** The performance quality of the judge (small LM) could be dependent on how the evaluation criteria are established.
>
> **Response:**
> Thank you for raising this point. We agree that the performance of the small-LM judge is influenced by the evaluation criteria it is trained to approximate. This dependency is inherent to all evaluator-based approaches, including prompting-based LLM-as-a-Judge methods, which rely on the same rubric specification.
>
> In our work, the evaluation dimensions are not arbitrarily designed. We follow widely adopted criteria from prior literature, including Roscoe [1] and Socreval [2], and we use one-shot prompt templates to obtain LLM evaluation scores for five standard aspects on GSM8K, MATH, and GPQA. While the prompt wording is written by us, the underlying evaluation aspects are directly aligned with these established metrics.
>
> We do observe variation in the difficulty of different criteria, but across all evaluation dimensions, the representation-based judge consistently outperforms prompting-based evaluation and supervised fine-tuning. In the revision, we will clarify this dependency and explicitly describe the origins of the evaluation criteria used in our experiments.
>
> **[1]** Golovneva, Olga, Moya Peng Chen, Spencer Poff, Martin Corredor, Luke Zettlemoyer, Maryam Fazel-Zarandi, and Asli Celikyilmaz. _ROSCOE: A Suite of Metrics for Scoring Step-by-Step Reasoning_. In _Proceedings of the Eleventh International Conference on Learning Representations (ICLR), 2024.
>
> **[2]** He, Hangfeng, Hongming Zhang, and Dan Roth. _Socreval: Large Language Models with the Socratic Method for Reference-Free Reasoning Evaluation_. In _Findings of the Association for Computational Linguistics: NAACL_, pages 2736–2764, 2024.

---

> ### Author Response · Authors · 2025-11-21
> **Response 2 to Reviewer Comments**
>
> **Questions:**
>
> **1.** Is there a specific reason why the paper focuses only on decoder-only models instead of considering encoder-only architectures?
>
> **Response:**
> Thank you for the question. Our work focuses on decoder-only models because the Semantic Capability Asymmetry Hypothesis is defined for generative architectures. The hypothesis characterizes a mismatch between what a model encodes in its intermediate representations and what it can express through autoregressive decoding, which naturally requires a generative model. Encoder-only architectures do not perform generation and therefore do not instantiate the generation-side capability that our formulation seeks to analyze.
>
> For completeness, we also evaluated a representative encoder-only model (RoBERTa) by fully fine-tuning it for the evaluation task. Its performance was noticeably lower than that of decoder-only models in our setup. We will clarify this point and include the corresponding results in the revision.
>
> **2.** I believe the Semantic Capacity Asymmetry Hypothesis aligns with a well-established understanding in LLM research. It would be helpful to cite prior work that supports this perspective.
>
> **Response:**
> Thank you for the suggestion. The Semantic Capacity Asymmetry Hypothesis is consistent with a broad body of work showing that language models often encode semantic information in their internal representations more reliably than they express it through surface outputs. Probing studies have demonstrated that hidden states contain rich syntactic and semantic structure that is only partially reflected in predictions [1][2]. Recent analyses of latent knowledge further show that task-relevant information can remain present in intermediate representations even when generated outputs are unreliable or intentionally misled [3][4]. These findings support the intuition that semantic understanding can be accessed more easily from representations than from autoregressive generation.
>
> Our contribution is to bring this perspective into the evaluation setting. We show that the semantic signals required for judging model outputs can be directly extracted from intermediate representations, leading to a lightweight and interpretable Representation-as-a-Judge approach that complements existing prompting-based evaluators. We will refine the discussion in the revision and incorporate the above references to clarify how they motivate the hypothesis in the evaluation context.
>
> **[1]** S. Kadavath et al. _Language Models (Mostly) Know What They Know_. CoRR, 2022.
>
> **[2]** A. Rogers, O. Kovaleva, and A. Rumshisky. _A Primer in BERTology: What We Know About How BERT Works_. Transactions of the Association for Computational Linguistics (TACL), 2020.
>
> **[3]** C. Burns, H. Ye, D. Klein, and J. Steinhardt. _Discovering Latent Knowledge in Language Models Without Supervision_. ICLR, 2023.
>
> **[4]** A. Mallen, M. Brumley, J. Kharchenko, and N. Belrose. _Eliciting Latent Knowledge from “Quirky” Language Models_. First Conference on Language Modeling, 2024.

---

### Official Review · Reviewer_GC61 · 2025-10-31

**Soundness:** 3
**Presentation:** 3
**Contribution:** 3
**Rating:** 4
**Confidence:** 4

**Summary:**

This paper observes that small LMs, although with their limited generative ability, can still provide reliable evaluations within their hidden representations. The authors then propose the Semantic Capacity Asymmetry Hypothesis, which posits that evaluation requires significantly less semantic capacity than generation and can rely on intermediate representations. The authors propose Representation-as-a-Judge, and implement this idea through INSPECTOR, a probing-based framework that predicts aspect-level evaluation scores directly from small model representations. Experimental results demonstrate that INSPECTOR substantially outperforms prompting-based small LMs and closely approximates large model judges.

**Strengths:**

**Novel idea for LLM-as-a-Judge.**
The concept of semantic capacity asymmetry is creative and appealing. It provides a fresh lens for understanding evaluation-relevant signals in internal representations, suggesting that weak evaluation performance in small LMs may arise from surface-level generation limitations rather than an inherent lack of semantic understanding.

**Effective probing design.**
The paper designs probing mechanisms to capture evaluation-related signals from model representations. These probes are empirically shown to be effective and help our understanding of how evaluation-relevant signals work in hidden layers.

**Practical significance.**
Building on this idea, the work demonstrates that smaller open-source models can serve as reliable evaluators, substantially reducing cost while maintaining evaluative fidelity, which is practically meaningful.

**Weaknesses:**

**1. Missing intuition behind the probing design.**
The paper introduces probing experiments to support the idea of semantic capacity asymmetry, but the intuition for the probing setup is underexplained. It is unclear what specific representational property the probes aim to capture or why this design shows semantic capacity.

**2.Limited dataset coverage.**
The evaluation focuses on three reasoning benchmarks (GSM8K, MATH, and GPQA), all within the mathematics and science domain. Other tasks types, such as open-ended generation, summarization, dialogue and code generation are not considered here. Incorporating a broader range of domains and task formats would strengthen the generality of the proposed idea.

**3. Clarity and presentation.**
Several sections are dense and could be improved with additional figures or concrete examples. For instance, a figure to illustrate how linear probes are constructed.

**Questions:**

1. Can the authors provide a formal definition or metric for semantic capacity asymmetry beyond the conceptual description?

2. The work uses judgments from DeepSeek-V3 as gold labels for training and evaluation, but does not incorporate human annotations as ground truth. Have the authors considered comparing (a) human judgments, (b) large LLM-as-a-Judge outputs, and (c) the proposed Representation-as-a-Judge ouputs?

3. Some minor questions:

   (1) In Eq (6), the symbol $\mathcal{C}$ is missing, and the meaning of $\mathcal{F}$ is unclear.

   (2) In the ablation study shown in Figure 4, is the classifier fixed as Logistic Regression in the left plot (where pooling methods vary), and is mean pooling fixed in the right plot (where classifier methods vary)?

---

> ### Author Response · Authors · 2025-11-21
> **Response 1 to Reviewer Comments**
>
> We thank the reviewer for the thoughtful and encouraging evaluation. We appreciate the positive remarks on the semantic capacity asymmetry perspective, our probing design, and the practical significance of small-model evaluators. The reviewer’s points highlight useful areas for clarification, including the probing intuition, domain breadth, and presentation. We address each item in detail below.
>
> **Weaknesses:**
>
> **1. Missing intuition behind the probing design.** The paper introduces probing experiments to support the idea of semantic capacity asymmetry, but the intuition for the probing setup is underexplained. It is unclear what specific representational property the probes aim to capture or why this design shows semantic capacity.
>
> **Response:**
> To clarify the intuition behind our probing setup: a probe does not modify or enhance the small LM. It simply takes the hidden representations generated by the frozen model and trains a very small classifier (e.g., logistic regression) to detect whether these representations contain the semantic signal associated with the target evaluation aspect. Because the probe itself has extremely limited capacity, good probe performance indicates that the relevant evaluative information—such as signals of correctness, inconsistency, or factual error—is already encoded in the model’s internal representations.
>
> The semantic capacity asymmetry we highlight refers to the gap between **generation** and **discrimination**. A small LM may lack the capacity to generate a correct and coherent solution through decoding, yet its internal representations can still encode clear **evaluative cues** that distinguish good responses from bad ones. Probing reveals these latent discriminative signals, showing that evaluation requires substantially less semantic capacity than generation. This is precisely why small LMs can serve as effective representation-based judges even though their prompting-based generation is weak.
>
> **2.Limited dataset coverage.** The evaluation focuses on three reasoning benchmarks (GSM8K, MATH, and GPQA), all within the mathematics and science domain. Other tasks types, such as open-ended generation, summarization, dialogue and code generation are not considered here. Incorporating a broader range of domains and task formats would strengthen the generality of the proposed idea.
>
> **Response:**
> Thank you for the helpful suggestions regarding evaluating our method on broader and more general tasks. While our primary focus is on reasoning evaluation, we additionally evaluated our approach on the open-ended generation benchmark AlpacaEval 2.0, using the same experimental settings as the three reasoning benchmarks reported in the paper.
>
> **Multiclass Classification (score=1-5)** on AlpacaEval 2.0:
> | | Semantic Consistency | Logicality | Informativeness | Fluency | Factuality |
> |:---:|:---:|:---:|:---:|:---:|:---:|
> | RoBERTa | 7.56% | 8.08% | 8.32% | 11.69% | 8.08% |
> | Qwen3-0.6B (Tuning) | 14.29% | 11.11% | 9.93% | 10.61% | 12.70% |
> | Qwen3-0.6B (Prompt) | 22.40% | 11.11% | 16.94% | 11.82% | 8.89% |
> | Qwen3-0.6B (Probing) | 36.84% | 42.59% | 46.71% | 61.21% | 23.33% |
> | Llama-3.2-1B-Instruct (Prompt) | 12.50% | 13.33% | 8.37% | 16.97% | 0.20% |
> | Llama-3.2-1B-Instruct (Probing) | 35.71% | 55.56% | 46.08% | 61.21% | 28.57% |
> | Qwen3-1.7B (Prompt) | 7.14% | 20.11% | 22.70% | 25.19% | 20.00% |
> | Qwen3-1.7B (Probing) | 35.12% | 45.93% | 42.01% | 65.10% | 23.70% |
> | Llama-3.1-8B-Instruct (Prompt) | 14.92% | 17.99% | 18.34% | 23.38% | 14.29% |
> | Llama-3.1-8B-Instruct (Probing) | 34.69% | 53.70% | 45.90% | 57.32% | 22.96% |
>
> **Binary-Classification (high vs low quality)** on AlpacaEval 2.0:
> | | Semantic Consistency | Logicality | Informativeness | Fluency | Factuality |
> |:---:|:---:|:---:|:---:|:---:|:---:|
> | RoBERTa | 41.56% | 39.68% | 46.58% | 49.49% | 39.68% |
> | Qwen3-0.6B (Tuning) | 35.24% | 43.00% | 45.41% | 28.48% | 27.78% |
> | Qwen3-0.6B (Prompt) | 55.24% | 43.00% | 49.93% | 46.36% | 44.44% |
> | Qwen3-0.6B (Probing) | 61.42% | 55.56% | 62.76% | 91.06% | 55.56% |
> | Llama-3.2-1B-Instruct (Prompt) | 49.20% | 34.19% | 59.26% | 64.24% | 39.68% |
> | Llama-3.2-1B-Instruct (Probing) | 67.14% | 70.42% | 67.44% | 81.82% | 65.80% |
> | Qwen3-1.7B (Prompt) | 50.24% | 65.80% | 45.03% | 45.45% | 55.56% |
> | Qwen3-1.7B (Probing) | 67.14% | 75.93% | 70.71% | 91.06% | 77.78% |
> | Llama-3.1-8B-Instruct (Prompt) | 39.77% | 48.15% | 58.34% | 62.42% | 27.35% |
> | Llama-3.1-8B-Instruct (Probing) | 70.16% | 77.78% | 69.96% | 82.12% | 64.07% |
>
> The results demonstrate that our probing methods yield consistent improvements over other baselines and exhibit trends similar to those reported in our paper. This further supports the effectiveness and robustness of our approach on a broader range of domains and tasks, including reference-free open-ended benchmarks.

---

> ### Author Response · Authors · 2025-11-21
> **Response 2 to Reviewer Comments**
>
> **Weaknesses:**
>
> **3. Clarity and presentation.** Several sections are dense and could be improved with additional figures or concrete examples. For instance, a figure to illustrate how linear probes are constructed.
>
> **Response:**
> Thank you for the helpful suggestion regarding clarity. We agree that several parts of the presentation, including the description of the probing process, can be made clearer, and we will carefully review these sections to improve readability. In the revision, we will improve the writing, provide a more explicit step-by-step explanation of how the probes are constructed, and include a small illustrative case that compares (1) the prompting-based evaluation results from a large LM and a small LM, and (2) the prediction made by our representation-based judge. We believe these additions will improve clarity without requiring changes to the overall method.
>
> **Questions:**
>
> **1.** Can the authors provide a formal definition or metric for semantic capacity asymmetry beyond the conceptual description?
>
> **Response:**
> Semantic capacity asymmetry refers to a state where a model successfully encodes the semantic information required for a task within its intermediate representations, yet fails to express this information reliably in generated language, revealing a gap between internal understanding and what can be externalized through its autoregressive decoding process.
>
> **2.** The work uses judgments from DeepSeek-V3 as gold labels for training and evaluation, but does not incorporate human annotations as ground truth. Have the authors considered comparing (a) human judgments, (b) large LLM-as-a-Judge outputs, and (c) the proposed Representation-as-a-Judge ouputs?
>
> **Response:**
> We appreciate the reviewer’s suggestion to compare (a) human judgments, (b) large LLM-as-a-Judge outputs, and (c) Representation-as-a-Judge outputs. Prior work has shown that high-capability LLM judges often exhibit strong agreement with human preferences, and using LLM-based judgments as a proxy for human annotation has become a standard and cost-effective practice in recent evaluation frameworks. Since our goal is to investigate whether a small model can recover the judgments of a strong LLM evaluator, rather than to replicate human annotation itself, we adopt DeepSeek-V3 as the supervising judge and do not conduct additional human labeling.
>
> Our experiments directly compare (b) large LLM-as-a-Judge outputs with (c) Representation-as-a-Judge outputs, and the observed agreement indicates that intermediate representations contain sufficient semantic information to reconstruct the decisions of a strong evaluator.
>
> If the reviewer considers a limited human-annotation comparison necessary, we are happy to include an additional small-scale human study during the rebuttal period.
>
> **3.** Some minor questions:
>
> (1) In Eq (6), the symbol $\\mathcal{C}$ is missing, and the meaning of $\\mathcal{F}$ is unclear.
>
> **Response:**
> We have clarified the notation in Eq. (6) in the revised draft. In Eq. (6), we have corrected the notation to explicitly use the set $\\mathcal{C}$, which denotes the classifier hyperparameterizations we evaluate. The selection domain is now written as $\\mathcal{S} \\times \\mathcal{P} \\times \\mathcal{C}$, replacing the previous symbol $\\mathcal{F}$ whose meaning was unclear.
>
> (2) In the ablation study shown in Figure 4, is the classifier fixed as Logistic Regression in the left plot (where pooling methods vary), and is mean pooling fixed in the right plot (where classifier methods vary)?
>
> **Response:**
> For Figure 4, the left plot varies the pooling methods with Logistic Regression fixed as the classifier, and the right plot varies the classifier choice with mean pooling fixed. We will revise the text to make this experimental setup clearer.

---

### Author Response · Authors · 2025-12-04
**Revision Note**

Dear ACs and Reviewers,

We have uploaded a revised version with minor clarifications.

Sections 3.1 and 3.2 refine the explanation of the evaluation criteria and improve the phrasing of the intuition behind probing, and Section 3.3 improves the notation and description of Equation (6).

Section 4.1 adds brief training-time details, and the caption of Figure 4 provides clearer context for the experimental setup.

Section 6 adds further background citations and improves phrasing, while Section H includes AlpacaEval2 results and Section J adds an additional case analysis.

No substantive changes were introduced.

Regards,

The Authors

---

### Meta-Review · Area_Chair_o9TW · 2026-01-06

**Summary:**

This paper introduces the Semantic Capacity Asymmetry Hypothesis, proposing that the semantic capacity required for "evaluation" is significantly lower than for "generation" and is embedded within the hidden representations of even small models. To leverage this, the authors develop INSPECTOR, a Representation-as-a-Judge framework that trains probes on intermediate layer states using DeepSeek-V3 as an oracle. Experimental results demonstrate that this approach significantly outperforms prompting and SFT, achieving performance competitive with Large Model Judges.

**Reviewer Concerns:**

Addressed Concerns:
1. Generalization across Domains (Reviewers GC61, CaW1, DiLJ, DGMU): Reviewers initially noted a limited experimental focus on mathematics. In response, the authors incorporated additional evaluations on AlpacaEval, demonstrating that the "Representation-as-a-Judge" phenomenon persists in general domains and confirming the broad applicability of their findings.
2. Reliability and Cost of Teacher Judges (Reviewers GC61, DiLJ, DGMU): To address concerns regarding the accuracy of the ground-truth labels, the authors clarified that their primary objective is to bridge the gap between SLMs and strong LLM judges, the latter of which have been proven to align well with human preferences in prior literature. Regarding cost concerns, the authors highlighted that the low token pricing of DeepSeek-V3 ensures the scalability and affordability of full-dataset evaluations.
3. Writing Clarity and Formalism (Reviewers GC61, CaW1): In response to requests for more rigorous definitions, concrete examples, and broader citations of theoretical work, the authors have substantially revised the manuscript. These enhancements (including additional figures and formal definitions) have resolved the initial clarity issues.
4. Model Architecture Scope (Reviewer CaW1): While the original study focused on decoder-only models, the authors addressed concerns regarding architectural diversity by including experiments with RoBERTa. These results somehow validated the efficacy of the proposed method within encoder-only architectures, albeit on a smaller scale.

**Reviewer Scores:**

- Reviewer GC61 (Score: 4): Maintained 4.
- Reviewer CaW1 (Score: 6): Maintained 6.
- Reviewer DiLJ  (Score: 6): Maintained 6.
- Reviewer DGMU (Score: 6): Maintained 6.

---

### Decision · Program_Chairs · 2026-01-26

Accept (Poster)